# AARS2 ameliorates myocardial ischemia *via* fine-tuning PKM2-mediated metabolism

Zongwang Zhang[1], Lixia Zheng[1], Yang Chen[1], Yuanyuan Chen[1], Junjie Hou[2], Chenglu Xiao[2], Xiaojun Zhu[1], Shi-Min Zhao[3], Jing-Wei Xiong[1,2]*

[1]Beijing Key Laboratory of Cardiometabolic Molecular Medicine, Institute of Molecular Medicine, College of Future Technology, Academy for Advanced Interdisciplinary Studies, and State Key Laboratory of Natural and Biomimetic Drugs, Peking University, Beijing, China; [2]School of Basic Medical Sciences and The Second Affiliated Hospital, Jiangxi Medical College, Nanchang University, Nanchang, China; [3]Obstetrics and Gynecology Hospital of Fudan University, State Key Lab of Genetic Engineering, School of Life Sciences and Institutes of Biomedical Sciences, Fudan University, Shanghai, China

**\*For correspondence:**
jingwei_xiong@pku.edu.cn

**Competing interest:** The authors declare that no competing interests exist.

## eLife Assessment

This **important** study highlights the essential role of AARS2 in safeguarding cardiomyocytes against ischemic stress by modulating energy metabolism towards glycolysis via PKM2. This mechanism unveils a promising new therapeutic target for treating myocardial infarction. **Convincing** findings are underpinned by a comprehensive dataset, including cardiomyocyte-specific genetic modifications, functional assays, and ribosome profiling, all collectively providing strong evidence for the critical involvement of the AARS2-PKM2 signalling pathway in cardiac protection.

**Abstract** AARS2, an alanyl-tRNA synthase, is essential for protein translation, but its function in mouse hearts is not fully addressed. Here, we found that cardiomyocyte-specific deletion of mouse AARS2 exhibited evident cardiomyopathy with impaired cardiac function, notable cardiac fibrosis, and cardiomyocyte apoptosis. Cardiomyocyte-specific AARS2 overexpression in mice improved cardiac function and reduced cardiac fibrosis after myocardial infarction (MI), without affecting cardiomyocyte proliferation and coronary angiogenesis. Mechanistically, AARS2 overexpression suppressed cardiomyocyte apoptosis and mitochondrial reactive oxide species production, and changed cellular metabolism from oxidative phosphorylation toward glycolysis in cardiomyocytes, thus leading to cardiomyocyte survival from ischemia and hypoxia stress. Ribo-Seq revealed that *Aars2* overexpression increased pyruvate kinase M2 (PKM2) protein translation and the ratio of PKM2 dimers to tetramers that promote glycolysis. Additionally, PKM2 activator TEPP-46 reversed cardiomyocyte apoptosis and cardiac fibrosis caused by AARS2 deficiency. Thus, this study demonstrates that AARS2 plays an essential role in protecting cardiomyocytes from ischemic pressure *via* fine-tuning PKM2-mediated energy metabolism, and presents a novel cardiac protective AARS2-PKM2 signaling during the pathogenesis of MI.

## Introduction

Cardiovascular disease (CVD) remains the leading cause of death worldwide (*Mendis et al., 2015*). Myocardial infarction (MI), a major type of CVDs, is considered an epidemic, affecting ~1–2 percent of adults and posing a serious threat to human health and life (*Thygesen et al., 2018*). MI is due to coronary atherosclerosis and coronary heart disease, and MI patients can develop arrhythmia, shock, heart failure, and even death in a short time in severe cases (*Frangogiannis, 2015*; *Thygesen et al., 2018*). MI results in massive loss of cardiomyocytes and abnormal cardiac remodeling with severe inflammation and fibrosis (*Zhang et al., 2022*). However, it is incompletely understood how to prevent massive cardiomyocyte death and inhibit cardiac fibrosis in time to achieve cardiac repair. Therefore, the main goals of cardiovascular research are to improve the adverse consequences of MI by improving immune inflammation and cardiac remodeling, cardiomyocyte protection, cardiomyocyte regeneration, and avoiding the expansion of fibrosis and scars.

As the heart ages, the reduced integrity and increased risk of heart diseases such as myocardial ischemia frequently take place, ultimately leading to heart failure (*Strait and Lakatta, 2012*). Due to limited regenerative potential of mammalian cardiomyocytes, massive death of cardiomyocytes leads to a decrease in their numbers, causing myocardial remodeling, hypertrophy, and impaired cardiac function after MI (*Whelan et al., 2010*). Over the years, the most effective approach to reducing acute myocardial ischemic injury is to salvage surviving cardiomyocytes. During ischemia, oxygen supply is reduced, leading to a decrease in ATP generation (*Kalogeris et al., 2012*). The significant reduction in ATP levels during ischemia is associated with the development of irreversible changes in cardiomyocytes because the cells deplete their energy reserves and cannot maintain internal homeostasis (*Saraste, 1999*). To compensate for the reduced oxidative phosphorylation (OXPHOS) during ischemia, cardiomyocytes increase glycolysis to maintain homeostasis. Under extreme stress conditions, such as MI, cardiomyocytes experience toxic levels of $Ca^{2+}$ and reactive oxygen species (ROS). Mitochondrial electron transport chain (ETC) uncoupling leads to excessive ROS production, resulting in lipid and protein oxidation, widespread cell damage, and initiation of cardiomyocyte death (*Penna et al., 2009*). The detrimental effects of ROS include DNA and RNA damage, lipid peroxidation, protein oxidation, and functional impairments (*Schieber and Chandel, 2014*). Mitochondria play a vital role in various metabolic pathways (*Sissler et al., 2017*). Mitochondria protein mutations, which are involved in mitochondrial metabolism and directly affect the central functions of mitochondria, have been extensively studied for a long time (*Gorman et al., 2016*). On the other hand, mitochondrial dysfunction and mutations in different components of the mitochondrial translation machinery may indirectly impact ATP production (*Pearce et al., 2013*). These include mitochondrial aminoacyl-tRNA synthetases (mt-AARSs).

Mt-AARSs are a group of nuclei-coding enzymes that ensure the accurate translation of the genetic code by binding 20 amino acids to their homologous tRNA molecules (*Meyer-Schuman and Antonellis, 2017*; *Ognjenović and Simonović, 2018*; *Sissler et al., 2017*). Cytoplasmic AARS provides amino acid-tRNA binding for protein translation, and the corresponding mt-AARS are introduced into the mitochondrial matrix, fulfilling their typical role of transporting amino acids for tRNA molecules encoded by their mitochondrial genomes (mt-tRNA). The accuracy of mitochondrial protein synthesis depends on the coordination of nuclear-encoded mt-AARS and mitochondrial DNA-encoded tRNA (*Ognjenović and Simonović, 2018*). Mitochondrial dysfunction is known to preferentially affect tissues with high energy requirements, especially the brain, muscles, and heart. It is widely recognized that nearly all nuclear genes associated with mt-AARS are considered pathogenic genes responsible for mitochondrial diseases (*Euro et al., 2015*). Typically, mutations in the mitochondrial alanyl-tRNA synthetase (AARS2) gene cause infantile-onset cardiomyopathy and white matter brain disease from childhood to adulthood (*Nielsen et al., 2020*). *AARS2* is the gene encoding mitochondrial alanyl-tRNA synthetase, which is responsible for connecting specific tRNA with its cognate amino acid, alanine (*Zhang et al., 2023b*). Recent studies have shown that AARS2 contains proline-hydroxylated conserved amino acid sequences, and its stability is enhanced under hypoxic conditions (*Mao et al., 2024*). Mutations in AARS2 can cause mitochondrial dysfunction (*Dallabona et al., 2014*; *Vasilescu et al., 2018*), leading to a reduction in cellular energy production (*Fine et al., 2019*). However, the precise mechanisms by which AARS2 deficiency contributes to the development of cardiomyopathy in patients are not fully understood, and the potential of AARS2 as a therapeutic target for heart-related diseases remains to be explored.

One of the nodal points of energy metabolism is pyruvate kinase that catalyzes the conversion of phosphoenolpyruvate (PEP) to pyruvate, the last step of glycolysis (*Méndez-Lucas et al., 2017*). Mammalian pyruvate kinases (PKs) are encoded by two genes (PKLR and PKM), and each can generate two isoforms, respectively (PKL and PKR; PKM1 and PKM2) (*Wong et al., 2015*). Pyruvate kinase muscle isozyme 1 (PKM1) and pyruvate kinase muscle isozyme 2 (PKM2) are alternative splicing products of the PKM gene and express in various tissues (*Israelsen and Vander Heiden, 2015*). PKM1 is found in some differentiated adult tissues, such as heart, muscle, and brain. PKM2 is widely expressed and a predominant isoform in many adult cell types, including kidney tubular cells, intestinal epithelial cells, and lung epithelial cells (*Dayton et al., 2016a*). PKM1 has constitutively high catalytic activity, whereas PKM2 enzyme activity is subject to complex allosteric regulation (*Ikeda et al., 1997*), which allows cells to switch between glycolysis and biosynthesis (*Tang et al., 2023*). PKM2 was recently found to protect cardiomyocytes from ischemia and promote cardiomyocyte proliferation (*Magadum et al., 2020*; *Ni et al., 2022*; *Wu et al., 2021*). However, it remains to be addressed how PKM2 protein and mRNA are regulated during pathogenesis of MI.

We have previously reported that AARS2 binds lactate to modulate metabolic proteins in ischemia hearts and skeletal muscles (*Du et al., 2022*; *Mao et al., 2024*). However, we have very limited knowledge on AARS2 function in the heart. And our previous experiments demonstrated that AARS2 overexpression had no significant effect on pyruvate dehydrogenase PDHA1 lactylation in cardiomyocytes. Here, we investigated AARS2 function under ischemic and hypoxic stress by employing both loss-of-function and gain-of-function of AARS2 in mice and neonatal rat cardiomyocytes (NRCMs). And we deciphered the mechanistic insights of AARS2 into energy metabolism by using a variety of cell biology, biochemistry, mass spectrometry, and transcriptomics technologies. This work not only reveals a novel cardiac protection signaling AARS2-PKM2, but also identifies therapeutic targets and small molecules for cardiomyopathy and MI.

## Results

### Deleting *Aars2* in adult cardiomyocytes causes heart failure

It is known that AARS2 deficiency causes human cardiomyopathy (*Nielsen et al., 2020*; *Vasilescu et al., 2018*), but the underlying mechanisms remain unclear. By comparing expression pattern of AARS2 before and after mouse MI, we found that both AARS2 proteins and mRNA decreased in the hearts after MI (*Figure 1A and B*), suggesting that AARS2 might be involved in the pathological progression of MI. To elucidate AARS2 function in adult cardiomyocytes, we crossed *Aars2*$^{floxed/}$$^{floxed}$ (*Aars2*$^{fl/fl}$) mice with α-*MHC-MerCreMer* mice to achieve cardiomyocyte-specific knockout of *Aars2* after tamoxifen treatment (*Figure 1C*). We used *Aars2*$^{fl/fl}$ mice as a control group, and α-*MHC-MerCreMer; Aars2*$^{fl/fl}$ mice as *Aars2* conditioned knockout (cKO) group (*Figure 1—figure supplement 1A*). As expected, *Aars2* cKO hearts had lower levels of AARS2 than that of *Aars2*$^{fl/fl}$ hearts (*Figure 1D*), while the levels of AARS2 were comparable in the liver, lung, and skeletal muscle between *Aars2*$^{fl/fl}$ and *Aars2* cKO groups (*Figure 1—figure supplement 1B*). The results suggest that we succeeded in achieving myocardial-specific deletion of *Aars2* in adult cardiomyocytes.

We then asked whether *Aars2* cKO mice have abnormal heart function. By applying intraperitoneal injection of tamoxifen to induce Cre-LoxP mediated cardiac specific deletion of *Aars2* for 5 d, we measured cardiac function by echocardiography (ECHO) at various time points and myocardial fibrosis by Masson staining at the experimental endpoint (*Figure 1E*). ECHO showed that ejection fraction (EF) and fractional shortening (FS) were comparable between *Aars2*$^{fl/fl}$ control and *Aars2* cKO mice before tamoxifen induction. At 7 d after tamoxifen induction, *Aars2* cKO mice showed reduced EF and FS, with a downward trend on days 14 and 28 after induction, whereas *Aars2*$^{fl/fl}$ control mice exhibited normal EF and FS (*Figure 1G*). Moreover, the heart wave tracings by ECHO were flatter in *Aars2* cKO mice than those of *Aars2*$^{fl/fl}$ mice at day 28 (*Figure 1F*). In addition, the *Aars2* cKO mice performed worse in terms of cardiac output, as well as left ventricle end-diastolic diameter (LVEDD), and left ventricle end-systolic diameter (LVESD) (*Figure 1H and I*). At day 28 after induction, 50% of *Aars2* cKO mice died, while all *Aars2*$^{fl/fl}$ mice survived (*Figure 1J*). Wheat germ agglutinin (WGA) staining showed that cardiomyocyte hypertrophy was evident in *Aars2* cKO hearts (*Figure 1K*), and Masson staining indicated severe cardiac fibrosis in *Aars2* cKO hearts compared with *Aars2*$^{fl/fl}$ hearts (*Figure 1L*) at 28 d after induction. The results collectively indicate that the deletion of *Aars2* in

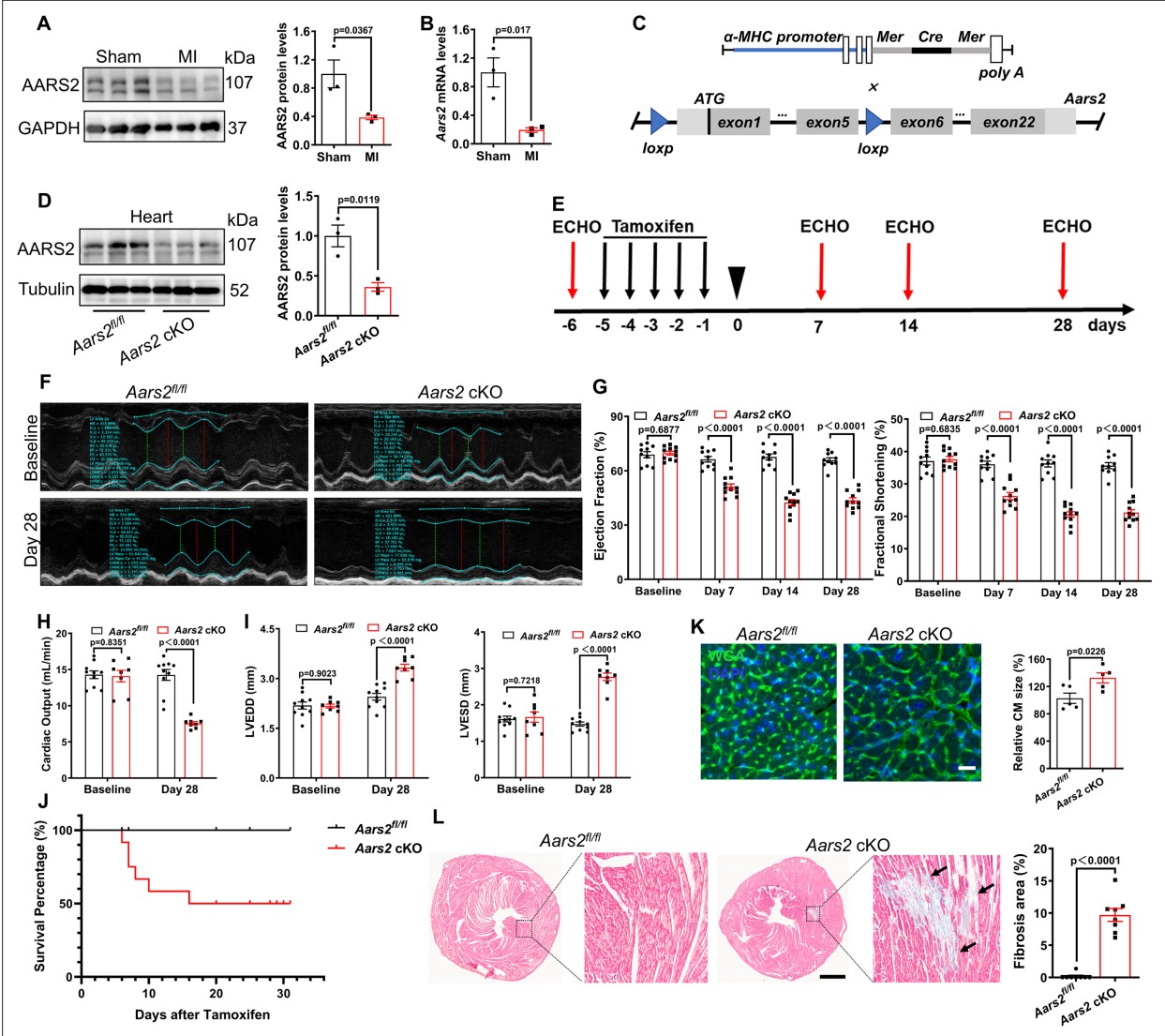

**Figure 1.** Cardiomyocyte-specific knockout of alanyl-tRNA synthetase (AARS2) leads to cardiac dysfunction and fibrosis in mice. (**A, B**) Western blot and quantitative real-time PCR (qRT-PCR) analysis showing reduced expression of AARS2 proteins (**A**) or mRNA (**B**) of 3 d myocardial infarction (MI) hearts compared with sham hearts (n=3). (**C**) Construction diagram of α-*MHC-MerCreMer* (upper) and *Aars2*^fl/fl mice (lower). (**D**) Western blots showing reduced AARS2 proteins in *Aars2* cKO hearts compared with *Aars2*^fl/fl hearts (n=3). (**E**) Schematic timelines of tamoxifen treatment and echocardiography (ECHO). (**F**) Representative M-mode tracings of ECHO in control and conditioned knockout (cKO) hearts before and after tamoxifen treatment. (**G**) Ejection fraction (EF) and fractional shortening (FS) of *Aars2*^fl/fl and *Aars2* cKO hearts at different time points after tamoxifen induction (n=10–11). (**H**) Cardiac output of *Aars2*^fl/fl and *Aars2* cKO mice at 28 d after tamoxifen induction (n=8–10). (**I**) Left ventricular end-diastolic diameter (LVEDD) and left ventricular end-systolic diameter (LVESD) in *Aars2*^fl/fl and *Aars2* cKO hearts at 28 d after tamoxifen induction (n=8–10). (**J**) Survival percentage of *Aars2*^fl/fl and *Aars2* cKO mice at 28 d after tamoxifen treatment (n=8–10). (**K**) WGA immunofluorescence showing cardiomyocyte hypertrophy on heart slices of *Aars2* cKO group compared with *Aars2*^fl/fl group at 28 d after tamoxifen induction (scale bar, 100 μm; n=5). (**L**) Masson staining and quantitative analysis showing increased cardiac fibrosis in *Aars2* cKO hearts compared with *Aars2*^fl/fl control hearts at 28 d after tamoxifen induction (scale bar, 1 mm; n=8). Mean ± s.e.m.

The online version of this article includes the following source data and figure supplement(s) for figure 1:

**Source data 1.** PDF file containing original western blots for *Figure 1A and D*.

**Source data 2.** Original files for western blot analysis displayed in *Figure 1A and D*.

**Figure supplement 1.** Genotyping of cardiomyocyte-specific *Aars2* knockout mice.

**Figure supplement 1—source data 1.** PDF file containing original western blots for *Figure 1—figure supplement 1A and B*.

**Figure supplement 1—source data 2.** Original files for western blot analysis displayed in *Figure 1—figure supplement 1A and B*.

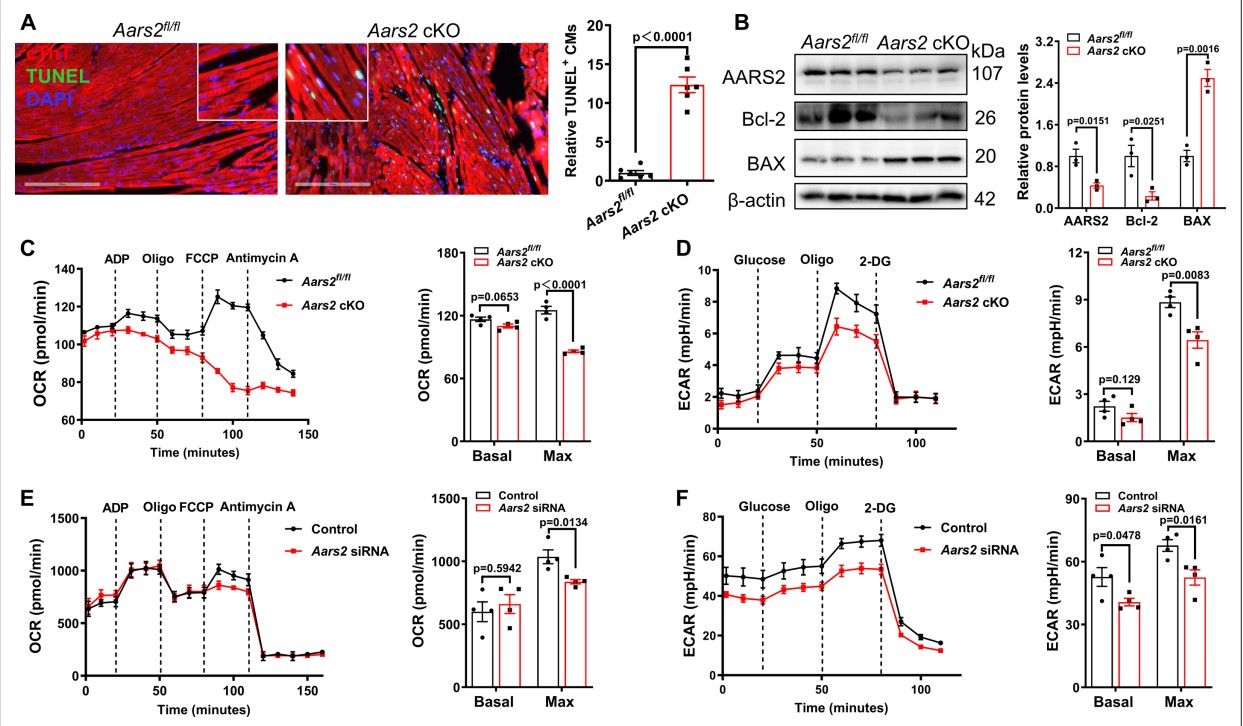

**Figure 2.** Cardiomyocyte-specific conditioned knockout (cKO) of alanyl-tRNA synthetase (AARS2) results in cardiomyocyte apoptosis and energy metabolism deficiency. (A, B) Immunofluorescence staining showing increased numbers of cTnT⁺ TUNEL⁺ cardiomyocytes (A) and Western blot showing reduced anti-apoptotic protein Bcl-2 and increased pro-apoptotic protein BAX (B) in *Aars2* cKO hearts compared with *Aars2^fl/fl* control hearts at 28 d post-tamoxifen treatment (scale, 200 μm; n=6 for panel A; n=3 for panel B). (C) Seahorse analysis showing reduced oxygen consumption rate (OCR) of cardiac mitochondria in *Aars2* cKO hearts compared with *Aars2^fl/fl* control hearts at 28 d after tamoxifen induction (n=4). (D) Seahorse analysis showing reduced extracellular acidification rate (ECAR) of adult mouse cardiomyocytes in *Aars2* cKO hearts compared with *Aars2^fl/fl* control hearts at 28 d after tamoxifen induction (n=4). (E–F) Seahorse analysis showing reduced OCR (E) and ECAR (F) of NRCMs in AARS2 siRNA group compared with control group at 3 d after transfection (n=4). Mean ± s.e.m.

The online version of this article includes the following source data and figure supplement(s) for figure 2:

**Source data 1.** PDF file containing original western blots for *Figure 2B*.

**Source data 2.** Original files for western blot analysis displayed in *Figure 2B*.

**Figure supplement 1.** Evaluating *Aars2* siRNA for knockout efficiency of alanyl-tRNA synthetase (AARS2) proteins in neonatal rat cardiomyocytes (NRCMs).

**Figure supplement 1—source data 1.** PDF file containing original western blots for *Figure 2—figure supplement 1A and B*.

**Figure supplement 1—source data 2.** Original files for western blot analysis displayed in *Figure 2—figure supplement 1A and B*.

cardiomyocytes results in abnormal cardiac function and fibrosis in mice, displaying characteristics of cardiomyopathy.

## AARS2 is required for energy metabolism in cardiomyocytes

In order to clarify the underlying mechanisms in *Aars2* cKO hearts, we investigated cardiomyocyte apoptosis and metabolism. TUNEL staining showed that *Aars2* cKO hearts had a significant increase of TUNEL⁺ cardiomyocytes compared with those in *Aars2^fl/fl* control hearts (*Figure 2A*). Consistently, the anti-apoptotic protein Bcl-2 decreased while the pro-apoptotic protein BAX increased in *Aars2* cKO hearts (*Figure 2B*). Given that the depletion of *Aars2* causes cellular energy metabolism imbalance in human patients (*Zhou et al., 2019*), we asked if *Aars2* cKO mouse cardiomyocytes exhibit metabolic defects. We performed Seahorse assays to evaluate mitochondrial OXPHOS and cardiomyocyte glycolysis from adult mouse hearts, as well as the OXPHOS and glycolysis of NRCMs. We found that the baseline of mitochondrial oxygen consumption rate (OCR) in *Aars2* cKO cardiomyocytes was similar to that in *Aars2^fl/fl* control cardiomyocytes, but the maximal OCR decreased in mutant mitochondria (*Figure 2C*). Furthermore, the baseline of extracellular acidification rate (ECAR) in *Aars2*

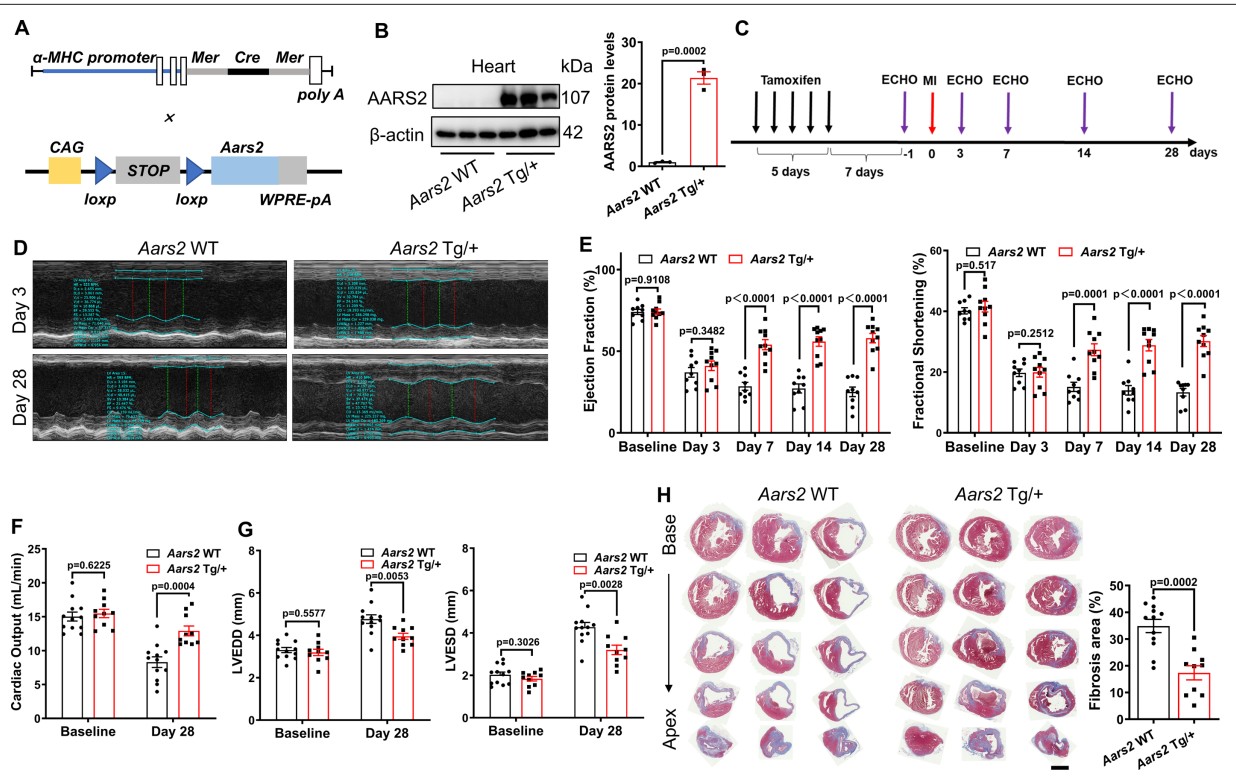

**Figure 3.** Cardiomyocyte-specific alanyl-tRNA synthetase (AARS2) overexpression improves cardiac function and decreases cardiac fibrosis in mice post-MI. (**A**) Schematic diagram of α-MHC-MerCreMer, and CAG-Aars2 mice that is driven by the CAG promoter. (**B**) Western blots showing transgenic overexpression of AARS2 proteins in the hearts of *Aars2* Tg/+compared with *Aars2* WT control mice (n=3). (**C**) Experimental protocols for Tamoxifen induction for 5 d, and then recovery for 7 d before echocardiography (ECHO) and myocardial infarction (MI). (**D**) Representative M-mode of ECHO in control and *Aars2* Tg/+mouse hearts at 3 d or 28 d post-MI. (**E**) Ejection fraction (EF) and fractional shortening (FS) of the *Aars2* WT and *Aars2* Tg/+mouse hearts were measured at different time points before and after MI (n=10–11). (**F**) The cardiac output of *Aars2* WT and *Aars2* Tg/+mice was measured before MI and 28 d after MI (n=10–11). (**G**) Left ventricle end-diastolic diameter (LVEDD) and left ventricle end-systolic diameter (LVESD) of *Aars2* WT and *Aars2* Tg/+ mice before MI and 28 d after MI (n=10–11). (**H**) Masson's staining showing decreased fibrotic area in the hearts of *Aars2* Tg/+compared with *Aars2* WT mice at 28 d after MI (scale bar, 1 mm, n=10–11). Mean ± s.e.m.

The online version of this article includes the following source data and figure supplement(s) for figure 3:

**Source data 1.** PDF file containing original western blots for *Figure 3B*.

**Source data 2.** Original files for western blot analysis displayed in *Figure 3B*.

**Figure supplement 1.** Cardiomyocyte-specific overexpression of alanyl-tRNA synthetase (AARS2) in the heart but not in the liver, lung, and skeletal muscle.

**Figure supplement 1—source data 1.** PDF file containing original western blots for *Figure 3—figure supplement 1A and B*.

**Figure supplement 1—source data 2.** Original files for western blot analysis displayed in *Figure 3—figure supplement 1A and B*.

**Figure supplement 2.** Overexpression of alanyl-tRNA synthetase (AARS2) in cardiomyocytes has no apparent effect on cardiomyocyte proliferation, hypertrophy, and angiogenesis after myocardial infarction (MI).

cKO cardiomyocytes was also similar to that of *Aars2*<sup>fl/fl</sup> control cardiomyocytes, but the maximal ECAR decreased in mutant cardiomyocytes (*Figure 2D*). In addition, knocking down *Aars2* by small interfering RNA (siRNA) in NRCMs (*Figure 2—figure supplement 1A and B*) resulted in similar effect on both maximal OCR and ECAR in vitro (*Figure 2E and F*). Taken together, these results indicate that depletion of *Aars2* in cardiomyocytes leads to cardiomyocyte death and impairs energy metabolism.

## Overexpression of AARS2 in cardiomyocytes alleviates MI in mice

To further elucidate the function of AARS2 in the pathology of MI, we generated cardiomyocyte-specific overexpression of AARS2 driven by the α-MHC promoter by crossing α-MHC-MerCreMer mice with *Aars2* transgenic mice (*Figure 3A*). In all experiments presented here, we used α-MHC-MerCreMer

(*Aars2* WT) mice as control group while used *α-MHC-MerCreMer*; *Aars2* Tg/+ (*Aars2* Tg/+) mice as experimental group (***Figure 3—figure supplement 1A***). Western blots showed that AARS2 was only overexpressed in the heart after tamoxifen induction for 5 d while normally expressed in the liver, lung, and skeletal muscle (***Figure 3B***, ***Figure 3—figure supplement 1***). In addition, immunofluorescence analysis revealed that AARS2 was highly expressed in cardiomyocytes of *Aars2* Tg/+ mouse hearts, whereas there was a significantly lower level of AARS2 in cardiomyocytes of *Aars2* WT control hearts (***Figure 3—figure supplement 1B***). Together, we have successfully generated transgenic mice with cardiomyocyte-specific overexpression of AARS2.

Then, we evaluated the impact of AARS2 overexpression on cardiac function post-MI by using *Aars2* transgenic mice (***Figure 3C***). Before MI, we confirmed that EF and FS were comparable between *Aars2* WT control and *Aars2* Tg/+ transgenic groups, and at day 3 after MI, EF and FS decreased significantly in both *Aars2* WT control and *Aars2* Tg/+ transgenic mouse groups. ECHO showed that the waveforms of *Aars2* Tg/+ transgenic hearts recovered well compared with those of *Aars2* WT control hearts at day 28 (***Figure 3D***). Importantly, from day 7–28 post-MI, we found that EF and FS gradually increased in *Aars2* Tg/+ transgenic group, while decreased in *Aars2* WT control group (***Figure 3E***). In addition, the cardiac output increased, while LVEDD and LVESD decreased in *Aars2* Tg/+ transgenic hearts compared with AATS2^{WT} control hearts at day 28 post-MI (***Figure 3F and G***). Consistently, Masson staining showed that cardiac fibrosis decreased in the *Aars2* Tg/+ transgenic hearts compared with *Aars2* WT control hearts at day 28 post-MI (***Figure 3H***). Briefly, these results suggest that overexpression of AARS2 in cardiomyocytes improves cardiac function and alleviates fibrosis in mice post-MI.

We next asked whether overexpression of AARS2 in cardiomyocytes has an impact on cardiomyocyte proliferation and coronary artery regeneration post-MI. We found that the proliferation index of either Ki67$^+$/cTnT$^+$ or pH3$^+$/cTnT$^+$ cardiomyocytes was comparable between *Aars2* Tg/+ transgenic and *Aars2* WT control hearts at 7 d post-MI (***Figure 3—figure supplement 2A and B***). Similarly, overexpression of AARS2 in NRCMs had no effect on either Ki67$^+$/cTnT$^+$ or pH3$^+$/cTnT$^+$

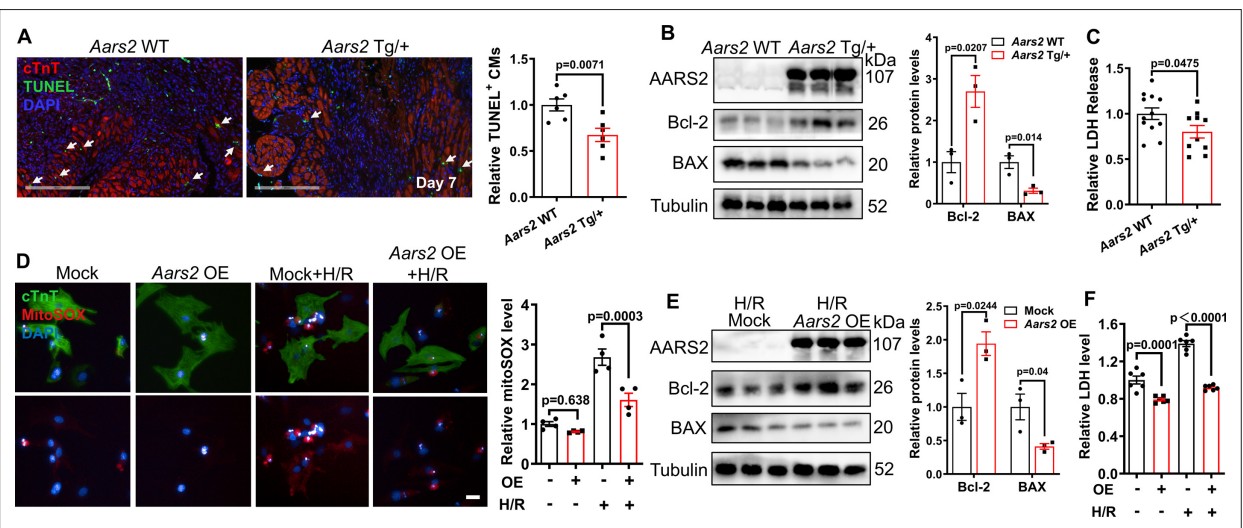

**Figure 4.** Overexpression of alanyl-tRNA synthetase (AARS2) attenuates cardiomyocyte apoptosis. (**A**) Immunofluorescence staining showing reduced cTnT$^+$/TUNEL$^+$ cardiomyocytes in *Aars2* Tg/+hearts compared with *Aars2* WT control hearts at 7 d after myocardial infarction (MI) (scale bar, 200 µm, n=6). (**B**) Western blots showing increased anti-apoptotic protein Bcl-2 and decreased pro-apoptotic protein BAX in *Aars2* Tg/+compared with *Aars2* WT control hearts at 7 d after MI (n=3). (**C**) The serum level of lactate dehydrogenase (LDH) decreased in *Aars2* Tg/+ hearts compared with *Aars2* WT control hearts at 28 d after MI (n=10). (**D**) Immunofluorescence staining and quantitative analysis showing reduced MitoSOX in *Aars2* OE neonatal rat cardiomyocytes (NRCMs) after 12 hr of hypoxia followed by 1 hr of reoxygenation (H/R, scale bar, 20 µm; n=4). (**E**) Western blots showing increased Bcl-2 and decreased BAX in NRCMs overexpressing *Aars2* (*Aars2* OE) compared with control NRCMs after 12 hr of hypoxia followed by 1 hr of reoxygenation (n=3). (**F**) The level of LDH decreased in *Aars2* OE NRCMs after 12 hr of hypoxia followed by 1 hr of reoxygenation (n=6). Mean ± s.e.m.

The online version of this article includes the following source data for figure 4:

**Source data 1.** PDF file containing original western blots for ***Figure 4B and E***.

**Source data 2.** Original files for western blot analysis displayed in ***Figure 4B and E***.

cardiomyocytes in vitro (*Figure 3—figure supplement 2C and D*). As compensatory collateral vessel formation in the infarcted area contributes to cardiac repair, we evaluated the coronary vessel density using anti-CD31 (an endothelial cell marker) and α-SMA (a smooth muscle cell marker). However, there were no changes in coronary vessel density between *Aars2* WT control and *Aars2* Tg/+ transgenic hearts at 7 d post-MI (*Figure 3—figure supplement 2E*). Additionally, WGA staining showed that AARS2 overexpression had no effect on cardiomyocyte hypertrophy in *Aars2* transgenic hearts (*Figure 3—figure supplement 2F*). In conclusion, these results suggest that overexpression of AARS2 has no direct influence on cardiomyocyte proliferation, hypertrophy, and arterial vessel formation during the pathology of MI.

## Overexpression of AARS2 decreases cardiomyocyte mtROS and apoptosis

Given that overexpression of AARS2 does not promote cardiomyocyte proliferation and hypertrophy in adult mice, we further investigate whether it salvages cardiomyocytes from ultimate death. TUNEL staining revealed less amounts of TUNEL$^+$/cTnT$^+$ cardiomyocytes in *Aars2* Tg/+ transgenic hearts compared with those in *Aars2* WT control hearts at 7 d post-MI (*Figure 4A*). Consistently, we found that the anti-apoptotic protein Bcl-2 increased, and pro-apoptotic protein BAX decreased in the left ventricles of *Aars2* Tg/+ transgenic hearts compared with *Aars2* WT control hearts (*Figure 4B*). The level of lactate dehydrogenase (LDH) in the blood serum is a standard marker for assessing the severity of MI, therefore, we examined LDH and found that it decreased in *Aars2* Tg/+ transgenic mouse serum (*Figure 4C*). In parallel, we also found that under hypoxia and reoxygenation (H/R), mitochondrial ROS indicator mtSOX decreased in *Aars2* overexpression (*Aars2* OE) NRCMs compared with Mock NRCMs

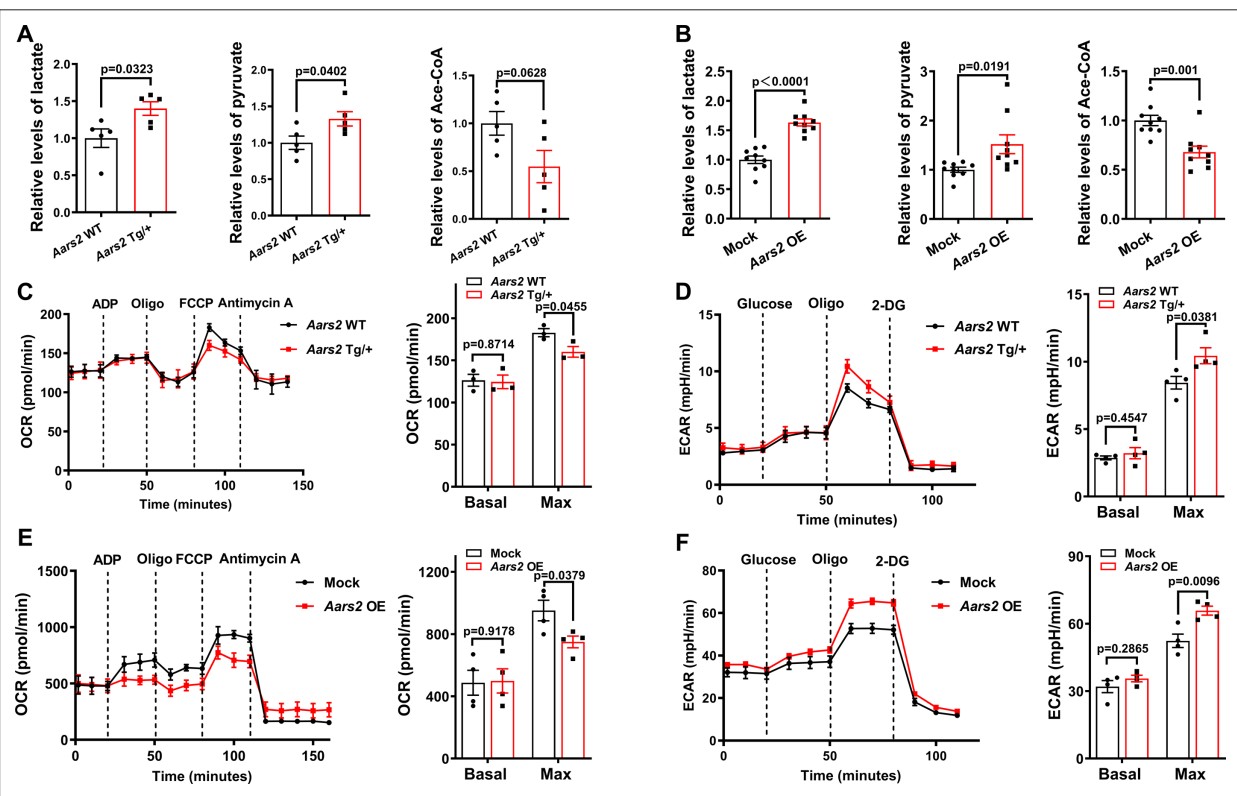

**Figure 5.** Cardiomyocyte overexpression of alanyl-tRNA synthetase (AARS2) regulates cardiac metabolism. (**A**) Mass spectrometry showing increased lactate and pyruvate but reduced acetyl-CoA in *Aars2* Tg/+ hearts compared with *Aars2* WT hearts after 7 d of myocardial infarction (MI) (n=5). (**B**) Mass spectrometry showing increased lactate and pyruvate but decreased acetyl-CoA in neonatal rat cardiomyocytes (NRCMs) overexpressing *Aars2* (*Aars2* OE) for 3 d (n=9). (**C**) Seahorse analysis showing oxygen consumption rate (OCR) of cardiac mitochondria in *Aars2* WT and *Aars2* Tg/+ mice at 28 ds after tamoxifen induction (n=3). (**D**) Seahorse analysis showing extracellular acidification rate (ECAR) and quantitative analysis of adult mouse cardiomyocytes in *Aars2* WT and *Aars2* Tg/+ mice at 28 d after tamoxifen induction (n=4). (E–F) Seahorse analysis showing OCR (**E**) and ECAR (**F**) of NRCMs in Mock control and *Aars2* OE groups at 3 d after transfection (n=4). Mean ± s.e.m.

(*Figure 4D*). Consistent with the findings in vivo, *Aars2* OE also decreased pro-apoptotic protein BAX while increased anti-apoptotic protein Bcl-2 in NRCMs subjected to H/R injury (*Figure 4E*), as well as decreased the release of LDH in the NRCM supernatants under either normoxia or hypoxia conditions (*Figure 4F*). Together, these results suggest that AARS2 overexpression alleviates ischemic- or hypoxia-induced cardiomyocyte damage.

## Overexpression of AARS2 in cardiomyocytes promotes glycolysis

We then asked what are the underlying mechanisms of AARS2 on cardiac protection. Since a previous study has shown that overexpression of AARS2 leads to intracellular metabolic changes (*Zhao et al., 2018*), we performed Mass Spectrometry for assessing metabolites of homogenates from heart tissues or NRCMs. We found that the amounts of lactate and pyruvate increased, while acetyl-CoA decreased in *Aars2* Tg/+ transgenic hearts compared with *Aars2* WT control hearts (*Figure 5A*). Similarly, we also found that AARS2 overexpression led to elevating lactate and pyruvate as well as decreasing acetyl-CoA in *Aars2* OE NRCMs compared with Mock control NRCMs (*Figure 5B*). To further assess the impact of AARS2 overexpression on cardiac energy metabolism, we conducted Seahorse to evaluate mitochondrial OXPHOS and cellular glycolysis in adult mouse hearts and NRCMs. The baseline of OCR was comparable between *Aars2* Tg/+ transgenic and *Aars2* WT control hearts, but the maximal OCR decreased in *Aars2* Tg/+ transgenic hearts (*Figure 5C*). Additionally, the baseline of ECAR had no difference in adult cardiomyocytes of both groups, but the maximal ECAR increased in *Aars2* Tg/+ transgenic cardiomyocytes compared with *Aars2* WT control cardiomyocytes (*Figure 5D*). We found similar findings that the baseline OCR and ECAR had no changes in both groups, but the maximal OCR decreased and maximal ECAR increased in *Aars2* OE NRCMs (*Figure 5E and F*). Thus, these results indicate that *Aars2* OE in cardiomyocytes can shift the metabolic profile from OXPHOS to glycolysis. Under ischemic or hypoxia conditions, this metabolic shift allows cardiomyocytes to avoid OXPHOS dysfunction and enables them to produce energy through glycolysis as a protective mechanism for self-preservation and energy supply. This metabolic adaptation may be crucial in maintaining cell viability and function during the periods of reduced oxygen availability in the heart.

## AARS2 promotes glycolysis *via* regulating the PKM2 translation and ratio of PKM2 dimers to tetramers

We then asked how elevated AARS2 in cardiomyocytes leads to metabolic changes from OXPHOS toward glycolysis. AARS2 is an alanyl-tRNA synthase that is critical for RNA translation, so we applied Ribosome RNA-seq to examine whether overexpression of AARS2 has any effect on RNA translation of metabolic genes. We found that AARS2 overexpression promoted the translation of glycolytic proteins, including PKM (*Figure 6A*), PDK4 and LDHA proteins (*Figure 6B*). Considering that PKM is the key enzyme that promotes glycolysis (*Tang et al., 2023*; *Vander Heiden et al., 2009*), which PKM1 and PKM2 are two different splicing forms of the PKM gene (*Dayton et al., 2016b*), we examined whether AARS2 overexpression has an effect on PKM1 and PKM2 translation. Western blots showed a significant increase in PKM2 proteins with AARS2 overexpression in both mouse hearts and NRCMs (*Figure 6D and E*). Conversely, PKM2 protein markedly decreased in *Aars2* cKO hearts (*Figure 6F*). On the other hand, we found that AARS2 overexpression had no effect on PKM1 proteins (*Figure 6C*), suggesting that AARS2 overexpression upregulates PKM2 but not PKM1 translation. In addition, overexpression of AARS2 also enhanced the translation of other signaling pathways including genes encoded by mitochondria, lipoprotein metabolic process, cellular response to hypoxia, and sodium ion transport (*Figure 6—figure supplement 1A–D*), which has a positive effect on mitochondrial function and cardiomyocyte function.

Because PKM2 can be converted between dimers and tetramers, which is critical for maintaining energy metabolism between glycolysis and OXPHOS, with dimers toward glycolysis (*Dayton et al., 2016b*). We found that AARS2 overexpression increased, while *Aars2* cKO decreased, PKM2 dimers and tetramers (*Figure 6G and H*), thus corroborating the increase of glycolysis in *Aars2* transgenic hearts and the reduction of glycolysis and OXPHOS in *Aars2* cKO. Previous studies have suggested that an increase in alanine causes a shift of PKM2 tetramers to dimers, consequently promoting glycolysis (*Dayton et al., 2016b*). Mass spectrometry analysis revealed that alanine indeed increased in transgenic hearts and NRCMs upon AARS2 overexpression (*Figure 6I*), which is consistent with the observation that the ratio of dimers/tetramers increased in *Aars2* transgenic hearts (*Figure 6J*). In

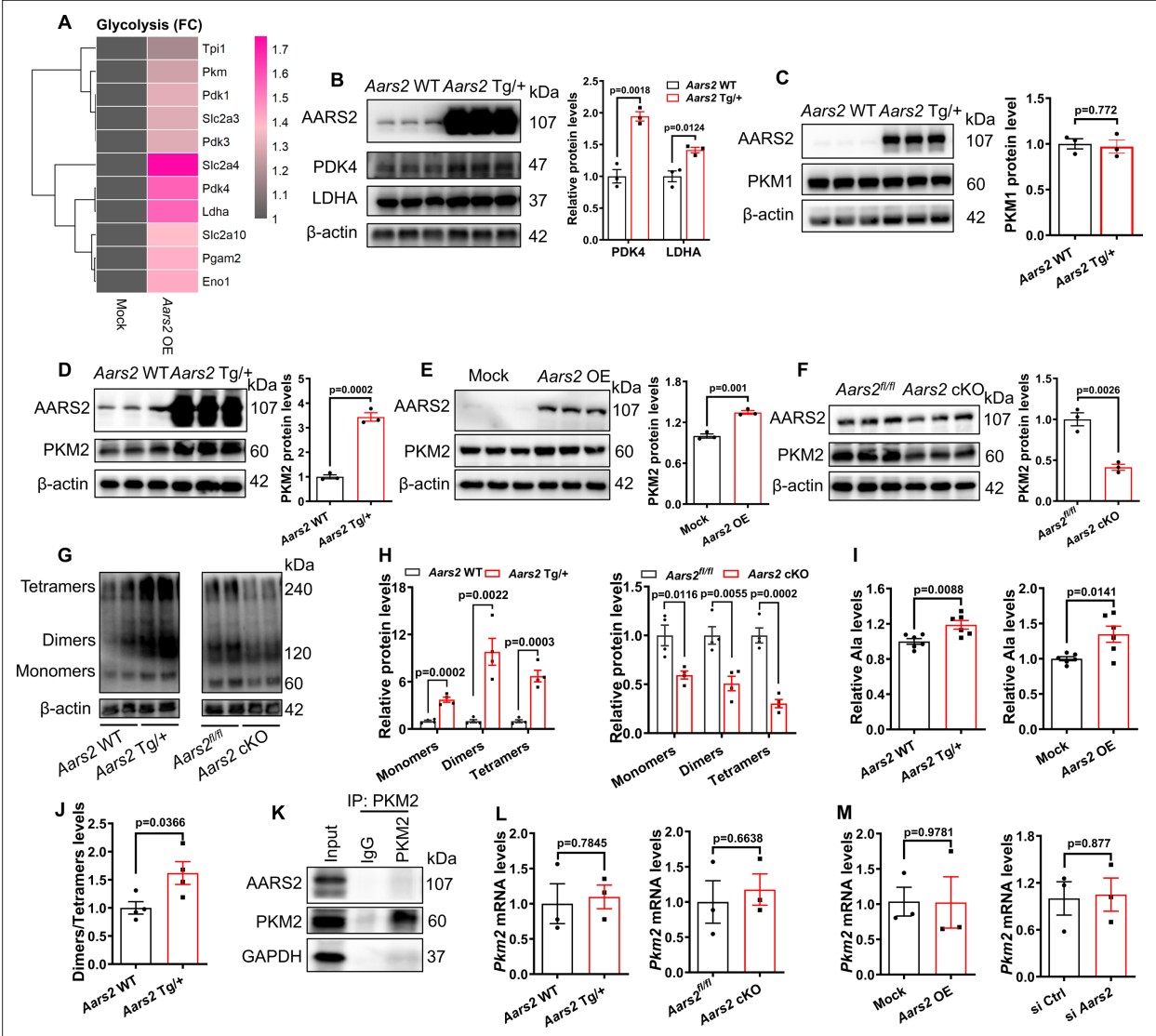

**Figure 6.** Overexpression of alanyl-tRNA synthetase (AARS2) increases the protein level of glycolytic pyruvate kinase M2 (PKM2) *via* enhancing PKM2 translation. (**A**) Ribosome RNA-seq showing elevated translation of signaling pathways of glycolysis in the *Aars2* OE NRCMs compared to the Mock neonatal rat cardiomyocytes (NRCMs). (**B**) Western Blots showing the level of AARS2, PDK4, and LDHA proteins in the hearts of *Aars2* WT control and *Aars2* Tg/+ transgenic mice (n=3). (**C**) Western Blots showing the level of AARS2 and PKM1 proteins in the hearts of *Aars2* WT control and *Aars2* Tg/+ transgenic mice (n=3). (**D–F**) Western Blots showing the level of AARS2 and PKM2 proteins in the hearts of *Aars2* WT control and *Aars2* Tg/+ transgenic mice (n=3) (**D**), in Mock control and *Aars2* OE NRCMs (**E**) (n=3), and in the hearts of *Aars2*fl/fl and *Aars2* cKO mice (**F**) (n=3). (**G–H**) Western Blots by non-denatured gels (**G**) and statistics (**H**) showing the amounts of PKM2 monomers, dimers, and tetramers in the hearts of *Aars2* WT control and *Aars2* Tg/+transgenic mice, and in the hearts of *Aars2*fl/fl and *Aars2* cKO mice (n=4). (**I**) Mass spectrometry analysis measuring the amounts of alanine (Ala) from homogenates of heart tissues (n=6) and NRCM lysates (n=6). (**J**) Ratio of quantitative results of PKM2 dimers and tetramers in the hearts of *Aars2* WT control and *Aars2* Tg/+ transgenic mice of panel H (n=4). (**K**) Co-immunoprecipitation reveals no evident interactions between PKM2 and AARS2 in NRCMs. (**L–M**) qRT-PCR showing the comparative level of *Pkm2* mRNA in the hearts of control sibling and *Aars2* Tg/+transgenic hearts; control sibling and *Aars2* cKO hearts (**L**); and in control, *Aars2* OE, and AARS2siRNA NRCMs (**M**) (n=3). FC, Fold changes; Mean ± s.e.m.

The online version of this article includes the following source data and figure supplement(s) for figure 6:

**Source data 1.** PDF file containing original western blots for *Figure 6B–G and K*.

**Source data 2.** Original files for western blot analysis displayed in *Figure 6B–G and K*.

**Figure supplement 1.** Overexpression of alanyl-tRNA synthetase (AARS2) increases the translation level of some cellular proteins.

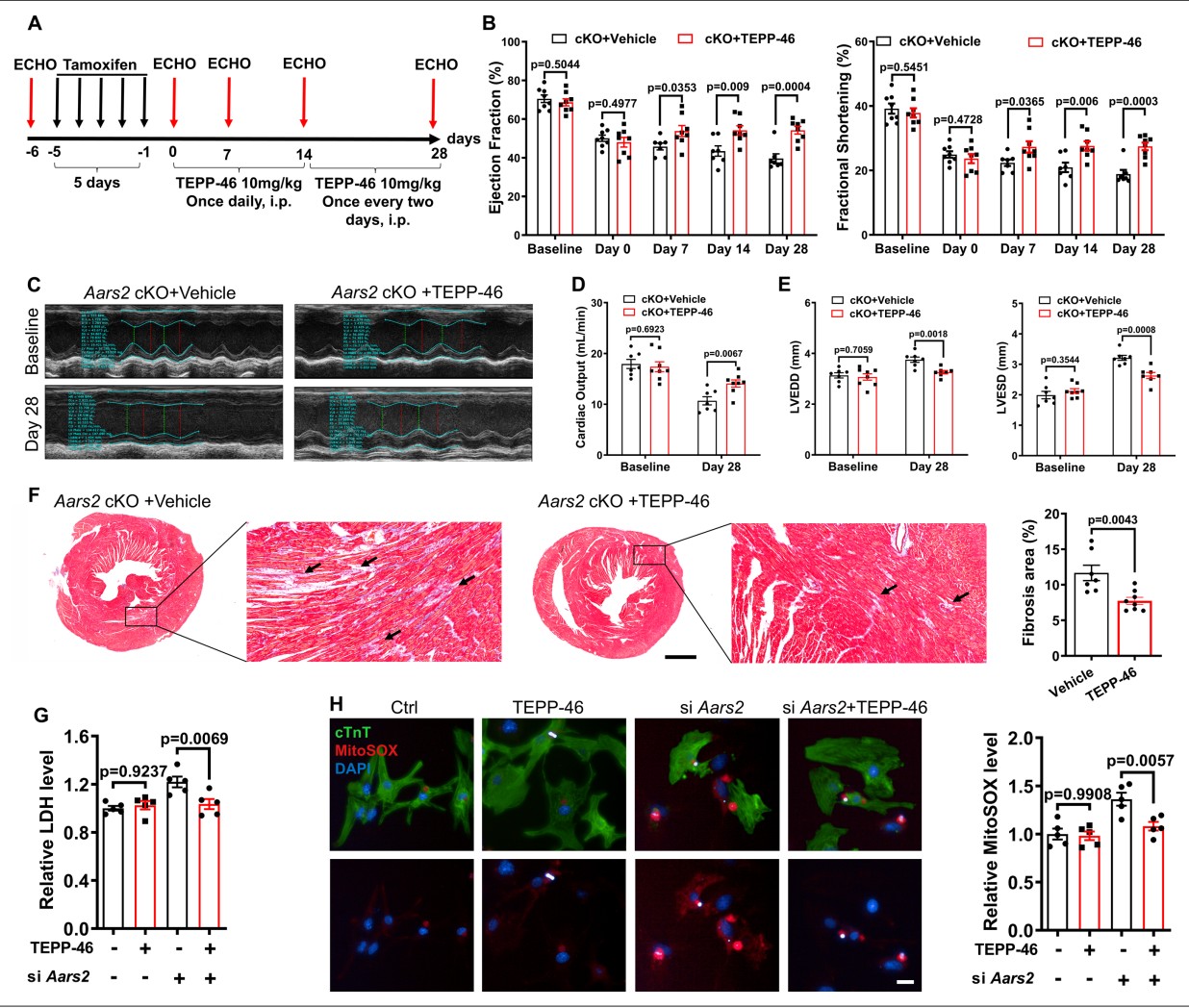

**Figure 7.** Pyruvate kinase M2 (PKM2) activator TEPP-46 improves cardiomyopathy in *Aars2* conditioned knockout (cKO) mice. (**A**) Experimental scheme and time points for tamoxifen induction, echocardiography (ECHO), and TEPP-46 administration. (**B**) Ejection fraction (EF) and fractional shortening (FS) of *Aars2* cKO mouse hearts at different time points after administration of control solvent and TEPP-46 (n=7–8). (**C**) Representative M-mode of ECHO in different groups of 4 wk mice. (**D**) Cardiac outputs of *Aars2* cKO mouse hearts at different time points after administration of control solvent and TEPP-46 (n=7–8). (**E**) Left ventricular end-diastolic diameter (LVEDD) and left ventricular end-systolic diameter (LVESD) at 28 d of *Aars2* cKO mice after administration of control solvent and TEPP-46 (n=7–8). (**F**) Masson staining showing cardiac fibrosis of *Aars2* cKO mouse hearts at 28 d after administration of either control solvent or TEPP-46 (scale bar, 1 mm; n=7–8). (**G**) Measurements of lactate dehydrogenase (LDH) release in neonatal rat cardiomyocytes (NRCMs) from the control group and TEPP-46 group (20 µM) after *Aars2* siRNA or control siRNA treatment for 72 hr (n=5). (**H**) Quantitative analysis of MitoSOX immunofluorescence in NRCMs from the control group and TEPP-46 group (20 µM) after *Aars2* siRNA or control siRNA treatment for 72 hr (scale bar, 20 µm; n=5). Mean ± s.e.m.

addition, co-immunoprecipitation assay revealed no direct interaction between AARS2 and PKM2 (*Figure 6K*). Moreover, regardless of *Aars2* overexpression or cKO in mouse hearts and NRCMs, we found that the level of *Pkm2* mRNA was not altered (*Figure 6L and M*), suggesting that AARS2 has no effect on PKM2 transcription. Therefore, AARS2 overexpression enhances glycolysis through regulating alanine and PKM2 translation.

## Activation of PKM2 alleviates cardiomyopathy in *Aars2* cKO mice

To establish the functional relationship between AARS2 and PKM2, we then investigated whether the PKM2 activator TEPP-46 (TEPP-46 tetramerizes PKM2, thereby enhancing energy supply) alleviates cardiomyocyte death caused by *Aars2* deficiency. After inducing *Aars2* deficiency with tamoxifen in *Aars2* cKO mice, we administered TEPP-46 according to the timeline and examined cardiac

function by ECHO as shown (*Figure 7A*). We found that the baseline of EF and FS were comparable before tamoxifen induction, EF and FS decreased at day 0 (5 d after tamoxifen induction) in both vehicle (*Aars2* cKO) control and TEPP-46 (*Aars2* cKO) groups. While EF and FS continued to decline in vehicle control mice, EF and FS exhibited a significant recovery and tended to stabilize in TEPP-46 mice after 7 d of treatment (*Figure 7B*). ECHO revealed better waveforms and an improvement in cardiac function in TEPP-46 group compared to vehicle group at 28 d post-treatment (*Figure 7C*). Moreover, increased cardiac output, decreased LVESD and LVEDD indicated a significant improvement in TEPP-46 group (*Figure 7D and E*). Masson staining showed less cardiac fibrosis in the hearts of TEPP-46 group compared with vehicle group at 28 d post-treatment (*Figure 7F*). Furthermore, *Aars2* knockdown in NRCMs resulted in a significant increase in LDH in vitro, which was repressed by TEPP-46 treatment (*Figure 7G*). Additionally, TEPP-46 also effectively suppressed mitochondrial ROS generation induced by *Aars2* knockdown (*Figure 7H*). Taken together, these results suggested that activating PKM2 by TEPP-46 effectively alleviates cardiomyopathy in *Aars2* deficient mice.

## Discussion

CVDs have become a prevalent cause of morbidity and mortality worldwide (*Zhang et al., 2023a*). The pathological processes of CVDs involve irreversible cardiomyocyte death, oxidative stress, inflammation, and cardiac fibrosis, which further contribute to adverse cardiac remodeling and ultimately lead to heart failure. Currently, there are no effective therapeutic strategies available for CVDs. This study showed that AARS2 plays a crucial role in salvaging cardiomyocytes from ischemic stress *via* fine-tuning PKM2-mediated energy metabolism, thus presenting novel therapeutic targets and potential small molecules for treating cardiomyopathy and MI.

Previous studies have shown that *AARS2* mutations lead to abnormal cardiomyocyte function, eventually to cardiomyopathy in human patients (*Nielsen et al., 2020*). However, it remains unknown whether AARS2 functions in cardiomyocytes or non-cardiomyocytes and how AARS2 regulates cardiomyocyte function and metabolism. Mitochondrial OXPHOS is the primary pathway for adult cardiomyocytes to produce ATP, relying on oxygen in the mitochondria to oxidize substrates and generate ATP through the cellular respiratory chain. Under hypoxia, glycolysis serves as an alternative pathway for cells to produce energy, breaking down glucose into lactate and generating a small amount of ATP (*Vander Heiden et al., 2009*; *Wei et al., 2023*). By creating cardiomyocyte-specific cKO and transgenic overexpression of *Aars2* mice, we found that AARS2 is not only required for cardiomyocyte function, but also improving cardiomyocytes from ischemia and hypoxia, thus establishing the critical function of AARS2 in cardiomyocytes. This provides a principle for future AARS2 gene therapy with targeted delivery into cardiomyocytes not non-cardiomyocytes. Our mechanistic investigations revealed that mitochondrial OXPHOS and glycolysis were impaired in cardiomyocytes lacking AARS2, leading to energy supply disruption and metabolic imbalance, ultimately resulting in cardiomyocyte death. On the other hand, AARS2 overexpression had a striking protective effect on cardiomyocytes, partly *via* enhancing glycolysis and resistance of cardiomyocytes to ischemia and hypoxia. Unbiased Ribosome RNA-Seq and functional analyses point out PKM2 as an important mediator of AARS2 in regulating cardiomyocyte function and metabolism. PKM2 decreased in the hearts of *Aars2* cKO mice, with the reduced formation of PKM2 dimers and tetramers; and AARS2 overexpression led to an increase in glycolysis and attenuated OXPHOS, with increased PKM2 translation and the ratio of PKM2 dimers to tetramers. PKM2 is a key enzyme in the glycolytic pathway, involved in converting PEP into pyruvate and ATP (*Méndez-Lucas et al., 2017*). Under certain conditions, an elevation in PKM2 levels can enhance the activity of the glycolytic pathway, thereby increasing the cell's ability to produce ATP through glycolysis (*Wong et al., 2015*). Recent studies have found that not only AARS1 (*Ju et al., 2024*; *Zong et al., 2024*), but also AARS2 is a lactate binding protein (*Mao et al., 2024*). Increased AARS2 further leads to increased binding to lactate, and PKM2 dimer, as a key enzyme in lactate production, can promote the production of lactate through glycolysis. Thus, AARS2 overexpression promotes the glycolytic pathway by increasing PKM2 translation and formation of dimers, thereby supporting the survival and function of cardiomyocytes. This metabolic adaptation is crucial for the heart to maintain its physiological functions under hypoxic or ischemic stress. Indeed, the PKM2 activator TEPP-46 enables to rescue cardiomyopathy phenotypes including cardiomyocyte death and cardiac fibrosis. This is because TEPP-46 enhances the tetramerization of PKM2, which

enhances OXPHOS for cardiomyocytes. Therefore, this work presents a new cardiomyocyte survival mechanism involving AARS2-PKM2 signaling under ischemic conditions.

It is widely recognized that cardiomyocytes undergo dramatic metabolic alterations and responses to oxidative stress during the pathogenesis of CVDs. Our Seahorse data suggested that the maximal OCR decreased, while the maximal ECAR strikingly increased, in *Aars2* Tg/+ transgenic hearts. These findings suggest an enhanced glycolytic pathway in AARS2-overexpressing cardiomyocytes, thereby augmenting their capacity for ATP production through glycolysis. This metabolic adaptation may contribute to sustaining the survival and functionality of cardiomyocytes, particularly under hypoxia or other stress conditions. This notion was also supported by functional analyses of AARS2 in NRCMs. Additionally, overexpression of AARS2 inhibits mtROS production and confers protection against ischemia and H/R-induced injury in cardiomyocytes. ROS represents a major oxidative species closely associated with cardiomyocyte damage and disease occurrence (*Wei et al., 2023*). Despite causing a reduction in OXPHOS, AARS2 overexpression also suppresses mtROS generation caused by ETC disorders, thus mitigating the detrimental effects of oxidative stress. Consequently, overexpression of AARS2 provides a protective mechanism by reducing mtROS levels and helps maintain normal physiological homeostasis within cardiomyocytes. In summary, upregulation or activation of PKM2 dimers mediated by AARS2 overexpression enhances glycolysis, thereby supporting the survival and functionality of cardiomyocytes, especially under hypoxic or stressful conditions. This metabolic adaptation coupled with reduction of ROS holds significant importance for maintaining normal cardiac function during challenging situations.

However, this study has certain limitations. The interaction between PKM2 and HIF-1α in the nucleus can affect the expression of glycolysis genes and promote cell survival (*Chen et al., 2014*; *Palsson-McDermott et al., 2015*; *Prakasam et al., 2018*), thus it remains be addressed whether AARS2 acts through this mechanism in cardiomyocytes. Investigating specific interactions between PKM2 and AARS2 in the context of cardiac pathophysiology will provide a better understanding of cardiomyocyte metabolism and adaptation mechanisms, but glycolysis is only one of the processes that the body makes adaptive changes under special conditions. It remains unclear whether continuously excessive high levels of glycolysis is beneficial to cardiomyocytes. This study primarily relies on mouse models and NRCMs, lacking sufficient human data to validate the experimental outcomes in human cardiomyocytes. Therefore, more researches related to human subjects are necessary before translating these findings to clinical treatments in humans. Moreover, ribosome profiling sequencing (Ribo-Seq) data show that AARS2 enhances PKM2 translation only as part of the investigated mechanisms. In fact, AARS2 also affects the translation of many other proteins. Further investigations are needed to address the influence of AARS2 on gene translation. Additionally, delving into more mutual interactions and regulatory mechanisms between AARS2 and PKM2 will help uncover the regulatory mechanisms of AARS2 on cardiomyocyte energy metabolism, providing a deeper understanding and potential therapeutic targets for the treatment of CVDs. Therefore, future investigations are needed to address how AARS2 regulates mitochondrial function in cardiomyocytes, how AARS2 interacts with PKM2 complex to regulate mitochondrial metabolism, and how the AARS2-PKM2 signaling is related to the pathogenesis of human cardiomyopathy.

# Materials and methods

**Key resources table**

| Reagent type (species) or resource | Designation | Source or reference | Identifiers | Additional information |
|---|---|---|---|---|
| Genetic reagent (*M. musculus*) | *Aars2*<sup>flox/flox</sup> | Zhao Lab, Fudan University | N/A | C57BL/6 J |
| Genetic reagent (*M. musculus*) | *Aars2* Transgenic | Zhao Lab, Fudan University | N/A | C57BL/6 J |
| Genetic reagent (*M. musculus*) | α-MHC-MerCreMer | Xiong Lab, Peking University | RRID:IMSR_GPT:T060079 | C57BL/6 J |
| Biological sample (rat) | Primary neonatal rat cardiomyocytes | Charles River | N/A | CD(SD) IGS Rat, 3 d |

*Continued on next page*

*Continued*

| Reagent type (species) or resource | Designation | Source or reference | Identifiers | Additional information |
|---|---|---|---|---|
| Antibody | anti-PKM2 (Rabbit polyclonal) | CST | 4053 S<br>RRID:AB_1904096 | WB (1:1000) |
| Antibody | anti-AARS2 (Rabbit polyclonal) | Abcam | ab197367<br>RRID:AB_2943036 | WB (1:1000) |
| Antibody | anti-Bcl2 (Rabbit polyclonal) | CST | 3498 S<br>RRID:AB_1903907 | WB (1:1000) |
| Antibody | anti-BAX (Rabbit polyclonal) | CST | 14796 S<br>RRID:AB_2716251 | WB (1:1000) |
| Antibody | anti-PKM1 (Rabbit polyclonal) | Proteintech | 15821–1-AP<br>RRID:AB_2163820 | WB (1:1000) |
| Antibody | anti-PDK4 (Rabbit polyclonal) | Bioss | bs-0682R<br>RRID:AB_10856420 | WB (1:1000) |
| Antibody | anti-LDHA (Rabbit polyclonal) | Bioss | bs-34202R<br>This paper | WB (1:1000) |
| Antibody | Goat Anti-Rabbit IgG (H&L)-HRP Conjugated | EASYBIO | BE0101-100<br>RRID:AB_3083002 | WB (1:10000) |
| Antibody | HRP-conjugated GAPDH | Proteintech | HRP-60004<br>RRID:AB_2737588 | WB (1:10000) |
| Antibody | HRP-conjugated Beta Actin | Proteintech | HRP-60008<br>RRID:AB_2819183 | WB (1:10000) |
| Antibody | HRP-conjugated Alpha Tubulin | Proteintech | HRP-66031<br>RRID:AB_2687491 | WB (1:10000) |
| Antibody | anti-Ki67 (Rabbit polyclonal) | CST | 12075 S<br>RRID:AB_2728830 | IF (1:300) |
| Antibody | anti-pH3 (Rabbit polyclonal) | CST | 53348 S<br>RRID:AB_2799431 | IF (1:300) |
| Antibody | anti-cTnT (Mouse polyclonal) | Abcam | ab8295<br>RRID:AB_306445 | IF (1:300) |
| Antibody | anti-α-SMA (Mouse polyclonal) | CST | 48938 S<br>RRID:AB_2799347 | IF (1:300) |
| Antibody | anti-CD31 (Rabbit polyclonal) | Abcam | ab182981<br>RRID:AB_2920881 | IF (1:300) |
| Antibody | Goat Anti-Rabbit IgG H&L (Alexa Fluor 488) | Abcam | ab150077<br>RRID:AB_2630356 | IF (1:500) |
| Antibody | Goat Anti-Mouse IgG H&L (Alexa Fluor 555) | Abcam | ab150118<br>RRID:AB_2714033 | IF (1:500) |

*Continued on next page*

*Continued*

| Reagent type (species) or resource | Designation | Source or reference | Identifiers | Additional information |
|---|---|---|---|---|
| Recombinant DNA reagent | pLenti-CMV-MCS-PGK-Puro-WPRE (plasmid) | This paper | N/A | CMV-MCS-PGK-Puro |
| Software, algorithm | Zen | Zeiss | RRID:SCR_013672 | https://www.zeiss.com/microscopy/en/products/software/zeiss-zen.html |
| Software, algorithm | GraphPad Prism | GraphPad | RRID:SCR_002798 | http://www.graphpad.com/ |
| Software, algorithm | Seahorse Wave | Agilent | RRID:SCR_014526 | http://www.agilent.com/en-us/products/cell-analysis-(seahorse)/software-download-for-wave-desktop |
| Software, algorithm | ImageJ | ImageJ | RRID:SCR_003070 | https://imagej.nih.gov/ij/ |

## Experimental animals

Sprague-Dawley neonatal rats (postnatal day 3, P3) and adult mice were obtained from Vital River Laboratory Animal Technology Co., Ltd (Beijing, China). All experimental procedures involving animal subjects were conducted in accordance with the protocols approved by the Institutional Animal Care and Use Committee at Peking University, Beijing, China (IACUC: IMM-XingJW-4).

Generation of cardiomyocyte-specific *Aars2* knockout mice and *Aars2* overexpression mice were achieved by crossing either *Aars2fl/fl* or *Aars2* Tg mice with *α-MHC-MerCreMer* transgenic mice (*Sohal et al., 2001*). The *Aars2fl/fl* mice and *Aars2* Tg mice were obtained from Dr. Shi-Min Zhao's lab at Fudan University, which were kept in C57BL/6 J background.

The *Aars2* cKO mice were given with TEPP-46 (10 mg/kg, MCE, Shanghai, China) by intraperitoneal injection once a day for the first and second weeks, and every other day for the third and fourth weeks after tamoxifen induction (20 mg/kg, Sigma-Aldrich, St. Louis, Missouri, USA).

## Adult mouse MI

The MI model was established in mice at an age ranging from 8–10 wk through ligation of the left anterior descending coronary artery (*Du et al., 2022*). Initially, each mouse was anesthetized by intraperitoneal injection of tribromoethanol (300 mg/kg; Sigma-Aldrich). Following complete anesthesia, the mouse was positioned supine and immobilized. Endotracheal intubation was performed, and the mouse was connected to a small-animal ventilator (MouseVent, Kent Scientific Corp., Torrington, CT, USA). The thoracotomy was carried out in the left intercostal space between the third and fourth ribs, exposing the heart by removing the pericardium. The left anterior descending coronary artery was permanently ligated using a 6–0 non-absorbable surgical suture, and the chest and skin were promptly sutured. Finally, the mouse was removed from the ventilator and kept warm until fully awaken.

## Echocardiography

ECHO was performed on anesthetized mice using a Vevo 3100 system (Visual Sonics, Toronto, ON, Canada) with 1.0% isoflurane anesthesia (*Zhang et al., 2016*). The hair on the left chest was completely removed using a depilatory paste to ensure optimal image acquisition. Two-dimensional ECHO images were obtained using a 20-MHz variable-frequency transducer in both the mid-ventricular short axis and the parasternal long axis. Heart rate, LVEDD, and LVESD were measured from the short-axis M-mode tracings at the level of the papillary muscle. Left ventricle functional parameters, including the percentage of EF, FS, and cardiac output were calculated using the aforementioned measurements and accompanying software. ECHO data were collected and analyzed on day –1 (baseline, 1 d before MI) as well as on days 3, 7, 14, and 28 post-MI.

## Pathological evaluation of mouse hearts

Mouse hearts were harvested and fixed in 4% paraformaldehyde (PFA) for a minimum of 2 d. Following fixation, the hearts underwent a series of processing steps, including dehydration, clearing, and infiltration, using a Histoprocessor (Tissue-Tek, Sakura, Tokyo, Japan). The paraffin-embedded tissues were then sectioned at a thickness of 5 μm. Masson's trichrome staining was performed according to

the protocol (*Chuang et al., 2016*). Briefly, heart sections were immersed in Bouin's solution, followed by sequential staining with Mayer's hematoxylin solution, Biebrich scarlet–acid fuchsin, phosphomolybdic acid–phosphotungstic acid, and aniline blue reagents (Sigma-Aldrich), with distilled water rinses between each step. Subsequently, the sections were dried, mounted on glass slides, and examined and photographed under a microscope.

## Isolation of NRCMs and transfection

NRCMs were isolated from P3 Sprague-Dawley rat hearts according to a previously described method (*Du et al., 2022*). Briefly, the hearts of rat pups were dissected and washed with $Ca^{2+}$- and $Mg^{2+}$-free HBSS (MacGene, Beijing, China). Using micro-dissecting scissors, ventricular tissues were minced into small pieces, and were then treated with 5 mL of digestion solution containing collagenase II (0.3 mg/mL; Thermo Fisher Scientific, Pittsburgh, PA, USA) and trypsin (1 mg/mL; Amresco, Pennsylvania, PA, USA) in HBSS for 5 min at 37 °C. The cell supernatant was collected and the residual tissue was repeatedly treated with the digestion solution until little remained. The supernatants were transferred to a tube with an equal volume of ice-cold Dulbecco's Modified Eagle Medium (DMEM; MacGene) containing 10% fetal bovine serum (FBS; Thermo Fisher Scientific, 16140071) and 1% penicillin-streptomycin, and then centrifuged at 400 g for 5 min. The cell pellets were re-suspended in 25 mL DMEM containing 10% FBS, 1% penicillin-streptomycin, and 1 µmol/L cytosine arabinoside. The cells were incubated in a 100 mm dish for 1.5 hr at 37 °C to eliminate fibroblast contamination, then non-adherent cells were collected and seeded at a concentration of $5\times10^5$ cells/mL. After incubation for 48–72 hr, the medium was removed and the NRCMs were then cultured with DMEM containing 10% FBS and 1% penicillin-streptomycin for further analysis.

Transfection of cells was performed by using Lipofectamine 3000 (Invitrogen) according to the manufacturer's protocol. Cells were harvested at 72 hr post-transfection and washed in phosphate-buffered saline for further RNA extraction and whole-cell protein extraction. The sequences of the small interfering RNA used are listed in *Supplementary file 1A*.

The vector pLenti-CMV-MCS-PGK-Puro-WPRE was selected for lentiviral packaging, and the *Aars2* gene was used according to the Gene Bank. Briefly, *Aars2* sequence was ligated with restriction enzymes. The vector was ligated, and the sequence-confirmed plasmids were transfected into 293T cells. The supernatant was collected, and lentiviruses at 10 MOI were used to infect the cells for AARS2 overexpression. AARS2 overexpression and its control, Ad-green fluorescent protein lentiviruses, were utilized. NRCMs were infected with the lentivirus for 72 hr. Then NRCMs were fixed for staining or lysed for protein extraction at 72 hr after removing virus from NRCMs.

## Genotyping

The last 1–3 mm of the mouse tails was used for genotyping. Lyophilized proteinase K (1245680100, Merck) was dissolved at 10 mg/mL in 50 mmol/L Tris-HCl, pH 8.0, with 5 mmol/L calcium acetate and stored at 4 °C. Then added 2 µL stock solution to tail tip samples that had been previously placed into 200 µL GNT-K buffer (10 mmol/L Tris-HCl, pH 8.5, 50 mmol/L KCl, 1.5 mmol/L $MgCl_2$, 0.01% gelatin, 0.45% Nonidet P-40, and 0.45% Tween20). The samples were digested for no more than 2 hr at 56 °C on a dry heat block. The samples were then heated at 95 °C for 15 min to inactivate proteinase K and were briefly centrifuged at 13,000 g for 1 min. Immediately after centrifugation, 2 µL DNA-containing supernatant was transferred to 20 µL reactions. 2x Taq Master Mix (Vayzme, P112-01) was used for the PCR reactions. The amplified products were then used for DNA electrophoresis. Primer pairs for genotyping are shown in *Supplementary file 1B*.

## Isolation of mitochondria from hearts

Mitochondria were isolated from mouse heart tissues as previously described (*Wieckowski et al., 2009*). In brief, mice were euthanized and heart tissue was immediately washed in ice-cold PBS (three times with PBS to remove the blood), then tissue was cut into small pieces using scissors. The used PBS solution was discarded and the tissue was washed once again with 10 mL of fresh, ice-cold IB-1. The heart tissue was transferred to an ice-cold glass/Teflon Potter Elvehjem homogenizer. IB-1 was added in the ratio of 4 mL of buffer per gram of heart tissue. Homogenization, as well as the following steps, were carried out at 4 °C to minimize the activation of proteases and phospholipases. After homogenization was completed, tissue was transferred to a centrifuge and spun at 700 g for 10 min

at 4 °C (repeated three times). The supernatant was collected and centrifuged at 9000 g for 10 min at 4 °C. The supernatant (representing the cytosolic fraction containing lysosomes and microsomes) was then discarded, and the pellet (containing the mitochondria) was gently resuspended in 20 mL of ice-cold IB-2. The mitochondrial suspension was then centrifuged at 10,000 g for 10 min at 4 °C (repeated two times). The pellet representing the crude mitochondrial fraction was used for further experimentation.

## Adult mouse cardiomyocyte isolation

Adult mouse cardiomyocytes were isolated according to a previously published method (*Li et al., 2021*). Briefly, after sacrifice, the mouse heart was immediately harvested and cannulated. The heart was then perfused with the modified Krebs-Ringer buffer (HEPES 20 mM, NaCl 137 mmol/L, KCl 5.4 mmol/L, MgCl$_2$·6H$_2$O 1.2 mmol/L, Na$_2$HPO$_4$ 1.2 mmol/L, glucose 10 mmol/L, taurine 10 mmol/L, pH 7.4) with collagenase II (0.8 mg/mL, Worthington, LS004177) and protease (0.04 mg/mL, Sigma-Aldrich, P5147). After digestion for approximately 25 min, atria were removed and ventricles were dissociated by pipetting. After filtration by cell strainer, adult cardiomyocytes were separated and enriched by sedimentation twice, and stored for further use.

## Seahorse metabolic assays

The OCR was measured using a Seahorse XFe24 analyzer (Agilent, USA). NRCMs were seeded at 50,000 cells/well onto XFe24 microplates (Agilent, 100777) in high glucose DMEM medium (Hyclone) containing 10% FBS, penicillin and streptomycin at 37°C and 5% CO$_2$. After 48 hr, cells were treated with lentivirus or transfected siRNA, and after 72 hr, seahorse assay medium was prepared by adding pyruvate, glutamine, and glucose to XF Base Medium. Cells were then cultured in the assay medium for 1 hr in a 37°C and non-CO$_2$ incubator prior to the assay. 50 µg of mitochondrial protein from the hearts was added to XFe24 microplates. A mitochondrial stress test kit (Agilent, 103015–100) was used to monitor OCR where baseline measurements were made followed by sequential injection of ADP (4 mmol/L), oligomycin (4 µmol/L), FCCP (2 µmol/L), and Rotenone (1 µmol/L). Data were analyzed in seahorse wave software. The ECAR was measured using a Seahorse XFe24 analyzer (Seahorse Bioscience) following the manufacturer's protocol. NRCMs were cultured and treated before assay following a similar protocol as in mitochondrial stress test assay. A glycolysis stress test kit (Agilent, 103020–100) was used to monitor ECAR where baseline measurements were made followed by sequential injection of glucose (10 mmol/L), oligomycin (1 µmol/L), 2-DG (50 mmol/L). Data were analyzed in Seahorse wave software.

## Hypoxia and reoxygenation assay

NRCMs were cultured for 48 hr under normoxia before being transferred to a hypoxia incubator with a gas mixture containing 37°C, 5% CO$_2$, and 0.1% O$_2$ balanced with nitrogen. After 12 hr of hypoxic culture, the cells were reoxygenated in the normal incubator containing 37°C, 5% CO$_2$ for 1 hr. Cells were immediately harvested at the indicated time points.

## Western blot analysis

Cells and left ventricular tissue were lysed with RIPA lysis buffer (Beyotime, P0013B, Shanghai, China) and protein concentrations were measured using the BCA assay kit (Beyotime, P0010). The total proteins were separated by sodium dodecyl sulfate-polyacrylamide gel electrophoresis and transferred to nitrocellulose membranes. After blocking, the membranes were incubated with anti-AARS2 (1:1000, Abcam, ab197367, Cambridge, UK), anti-B-cell lymphoma-2 (Bcl-2, 1:1000, CST, 3498 S), anti-Bcl-2 associated X (BAX, 1:1000, CST, 14796 S), anti-PKM2 (1:1000, CST, 4053 S), anti-PKM1 (1:1000, Proteintech, 15821–1-AP), anti-PDK4 (1:1000, Bioss, bs-0682R) and anti-LDHA (1:1000, Bioss, bs-34202R). After washing with TBS with Tween-20, Goat Anti-Rabbit IgG (H&L)-HRP Conjugated (1:10,000, EASYBIO, BE0101-100) was added, HRP-conjugated GAPDH Monoclonal antibody (1:10,000, Proteintech, HRP-60004), anti-β-actin (1:10,000, Proteintech, HRP-60008) and anti-α-Tubulin (1:10000, Proteintech, HRP-66031) were used as a control for normalization. Protein signals were detected with Super ECL Detection Reagent (Yeasen, 36208ES60) in a ChemiDoc MP Imaging System (Bio-Rad). Native proteins were extracted using Column Tissue & Cell Protein Extraction Kit (Epizyme, PC201 plus) according to the manufacturer's instructions.

## Co-immunoprecipitation

For immunoprecipitation, cells with an 85% confluent at 15 cm plate were used and equilibrated to serum-free DMEM for 2 hr prior to immunoprecipitation. The total protein from cell lysate was immunoprecipitated. First, the extract was incubated with anti-PKM2 for 24 hr at 4 °C. The protein A/G Agarose (Beyotime, P2012) was added and further incubated for 3 hr at 4 °C followed by centrifugation at 12,000 g for 5 min. Then recovered the precipitate for washing, resuspended it in 40 μL SDS lysis buffer and boiled for 5 min, and finally analyzed the precipitate by immunoblotting with the indicated antibody.

## Immunofluorescence cytochemistry

Heart tissues were embedded in OCT compound and sectioned at 6 μm on a cryostat. After fixation in PFA, sections were permeabilized with 1% Triton X-100 for 10 min and blocked with 1% BSA for 60 min. NRCMs on cell culture plates were fixed in 4% PFA for 30 min and permeabilized with 1% Triton X-100 for 10 min and blocked with 1% BSA for 1 hr. Then, slices and NRCMs were incubated with rabbit anti-AARS2 (1:300; Abcam, ab197367), rabbit anti-Ki67 (1:300; CST, 12075 S), rabbit anti-pH3 (1:300; CST, 53348 S), rabbit anti-CD31 (1:300; Abcam, ab182981), mouse Anti-Cardiac Troponin T (cTnT, 1:300; Abcam, ab8295), and mouse anti-α-SMA (1:300; CST, 48938 S) at 4 °C overnight. After washing with 1% PBS-Tween, Alexa Fluor 488-conjugated anti-rabbit and Alexa Fluor 555-conjugated anti-mouse secondary antibodies (1:500; Abcam, ab150077, ab150118) were added. Terminal deoxynucleotidyl transferase dUTP nick end labeling (TUNEL) assay was performed using the TUNEL Apoptosis Detection Kit (Alexa Fluor 488; Beyotime, C1088) according to the manufacturer's instructions. MitoSOX (Thermo Fisher Scientific, M36008) staining was applied directly on the NRCMs for 10 min following the manufacturer's instructions. All the sections were counterstained with 4', 6-diamidino-2-phenylindole (DAPI, Abcam, ab104139). Fluorescent cells and tissues were visualized and digital images were captured using Axio Scan Z1 (Carl Zeiss, Germany).

## WGA staining

To determine the cross-section areas of CMs, heart sections were dewaxed in xylene, rehydrated in an ethanol series, washed in PBS, and then incubated for 1 hr with Wheat Germ Agglutinin, Alexa Fluor 488 Conjugate (5 μg/mL, Invitrogen, W11261). Added mouse Anti-cTnT (1:300; Abcam, ab8295) at 4 °C overnight. Then added Alexa Fluor 555-conjugated anti-mouse secondary antibodies (1:500; Abcam, ab150118) for 1 hr. The slides were then rinsed in PBS and co-stained with DAPI. To quantify the size of cells, images at 20×xmagnification were analyzed using ImageJ.

## LDH release assay

The mice serum and cell lysate were collected and analyzed for LDH release using an LDH-release kit (Beyotime, C0016) following the manufacturer's instructions.

## Quantitative real-time PCR

mRNA was isolated from cells or tissues with a Total RNA kit (Tiangen, DP424, Shanghai, China), and reverse-transcribed with an RT Master Mix kit (Yeasen, 111141ES60, Shanghai, China). Quantitative Real-Time PCR (qRT-PCR) was performed with the SYBR Premix Ex Taq kit (Yeasen, 11184ES08) on an AB 7500 Fast Real-Time PCR System (Applied Biosystems). *β-Actin* and *Gapdh* were used as an internal control. The sequences of primers are summarized in *Supplementary file 1C*. The fold changes in mRNA expression levels were normalized to *β-Actin* and *Gapdh* using the $^{\Delta\Delta}$Ct method.

## Metabolite profiling

Frozen mouse heart tissues were homogenized with TissueLyser II by adding the −20°C-cold high-performance liquid chromatography-grade methanol to the final concentration of 100 mg/mL. The denatured protein was pelleted by centrifuging the tube at 14,000 rpm for 20 min at 4°C and the supernatant of individual sample was transferred into the new tube for metabolite profiling analysis. Before the LC-MS-based analysis, 10 μL of supernatant was diluted with 90 μL of methanol for positive mode analysis, and 30 μL of supernatant was diluted with 70 μL of acetonitrile/methanol (3:1) for negative mode analysis, respectively.

10 µL of reconstituted sample was loaded onto either a 150×2.1 mm Atlantis HILIC column (Waters, Milford, MA) for positive mode analysis or 5 µL was loaded on a 100×2.1 mm 3.5 µm XBridge amide column (Waters) for negative mode analysis using an HTS PAL Autosampler (Leap Technologies, Carrboro, NC) or Agilent 1290 Infinity Autosampler. The metabolites were separated using an Agilent 1200 Series high-performance liquid chromatography system (Agilent Technologies, Santa Clara, CA) coupled to a 4000-QTRAP mass spectrometer (AB SCIEX, Foster City, CA) in positive mode analysis; an Agilent 1290 infinity high-performance liquid chromatography binary pump system (Agilent Technologies) coupled to a 6490-QQQ mass spectrometer (Agilent Technologies) in negative mode analysis MultiQuant software v2.1 (AB SCIEX) and MassHunter quantitative software were used for automated peak integration, respectively, and metabolite peaks were also assessed manually for quality of peak integration.

## Ribosome profiling sequencing

Ribo-Seq of NRCMs was performed by Novogene. Briefly, use RNase I to digest the unprotected RNA, leaving only the ribosome-protected mRNA fragments, and sequencing libraries were constructed and carried out on an Illumina Novaseq 6000 SE50. Differential gene expression of control and *Aars2* OE groups was analyzed using the DESeq2 R package. DESeq2 provides statistical routines for determining digital differential gene expression data using a model based on the negative binomial distribution. We used the clusterProfiler R package to test the statistical enrichment of differentially expressed genes.

## Statistical analysis

Values are reported as mean ± s.e.m. unless indicated otherwise. The two-tailed non-parametric Student's t-test was used for two-group comparisons by Prism 9.0 (GraphPad Software, San Diego, CA, USA) statistical software. In addition, one-way analysis of variance analysis (ANOVA) was used to evaluate the statistical significance for multiple-group comparisons. Values of $p < 0.05$ were considered statistically significant.

## Acknowledgements

The authors thank the National Center for Protein Sciences at Peking University in Beijing; and the members of Dr. Jing-Wei Xiong's laboratory for helpful discussions and technical assistance. This work is supported by grants from the National Key R&D Program of China (SQ2023YFA1800026 and 2018YFA0800501 to JWX and XZ); and the National Natural Science Foundation of China (32230032, 31730061, and 81870198 to JWX and XZ).

## Additional information

### Funding

| Funder | Grant reference number | Author |
|---|---|---|
| National Key Research and Development Program of China | SQ2023YFA1800026 | Jing-Wei Xiong Xiaojun Zhu |
| National Key Research and Development Program of China | 2018YFA0800501 | Jing-Wei Xiong Xiaojun Zhu |
| National Natural Science Foundation of China | 32230032 | Jing-Wei Xiong Xiaojun Zhu |
| National Natural Science Foundation of China | 31730061 | Jing-Wei Xiong Xiaojun Zhu |
| National Natural Science Foundation of China | 81870198 | Jing-Wei Xiong Xiaojun Zhu |

| Funder | Grant reference number | Author |
|---|---|---|

The funders had no role in study design, data collection and interpretation, or the decision to submit the work for publication.

## Author contributions

Zongwang Zhang, Conceptualization, Data curation, Software, Formal analysis, Validation, Investigation, Methodology, Writing – original draft, Project administration, Writing – review and editing; Lixia Zheng, Yang Chen, Yuanyuan Chen, Junjie Hou, Chenglu Xiao, Shi-Min Zhao, Methodology; Xiaojun Zhu, Resources, Supervision, Project administration; Jing-Wei Xiong, Conceptualization, Resources, Supervision, Writing – original draft, Project administration, Writing – review and editing

## Author ORCIDs

Zongwang Zhang ⓘ https://orcid.org/0009-0008-3711-0090
Yang Chen ⓘ https://orcid.org/0009-0006-2087-8875
Jing-Wei Xiong ⓘ https://orcid.org/0000-0001-8438-4782

## Ethics

All experimental procedures involving animal subjects were conducted in accordance with the protocols approved by the Institutional Animal Care and Use Committee at Peking University, Beijing, China (IACUC: IMM-XingJW-4).

Reviewer #1 (Public review): https://doi.org/10.7554/eLife.99670.3.sa1
Reviewer #3 (Public review): https://doi.org/10.7554/eLife.99670.3.sa2
Author response https://doi.org/10.7554/eLife.99670.3.sa3

---

# Additional files

## Supplementary files

Supplementary file 1. *Aars2* siRNA sequences, genotyping primers, and real-time PCR primers for mouse and rat models.

MDAR checklist

## Data availability

RNA-Seq data have been deposited at ArrayExpress (E-MTAB-13767).

The following dataset was generated:

| Author(s) | Year | Dataset title | Dataset URL | Database and Identifier |
|---|---|---|---|---|
| Zhang Z | 2024 | Ribosome sequencing of the effect of control lentivirus and AARS2 overexpression lentivirus on neonatal rat cardiomyocyte translation | https://www.ebi.ac.uk/biostudies/ArrayExpress/studies/E-MTAB-13767?query=E-MTAB-13767 | ArrayExpress, E-MTAB-13767 |

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
