## [Editor Report · eLife Assessment]

This **important** study highlights the essential role of AARS2 in safeguarding cardiomyocytes against ischemic stress by modulating energy metabolism towards glycolysis via PKM2. This mechanism unveils a promising new therapeutic target for treating myocardial infarction. **Convincing** findings are underpinned by a comprehensive dataset, including cardiomyocyte-specific genetic modifications, functional assays, and ribosome profiling, all collectively providing strong evidence for the critical involvement of the AARS2-PKM2 signalling pathway in cardiac protection.

---

## [Referee Report · Reviewer #1 (Public review)]

In this study, the authors introduced an essential role of AARS2 in maintaining cardiac function. They also investigated the underlying mechanism that through regulating alanine and PKM2 translation are regulated by AARS2. Accordingly, a therapeutic strategy for cardiomyopathy and MI was provided.

Comments on revised version:

The authors have completely addressed my concerns.

---

## [Referee Report · Reviewer #3 (Public review)]

In the present study, the author revealed that cardiomyocyte-specific deletion of mouse AARS2 exhibited evident cardiomyopathy with impaired cardiac function, notable cardiac fibrosis, and cardiomyocyte apoptosis. Cardiomyocyte-specific AARS2 overexpression in mice improved cardiac function and reduced cardiac fibrosis after myocardial infarction (MI), without affecting cardiomyocyte proliferation and coronary angiogenesis. Mechanistically, AARS2 overexpression suppressed cardiomyocyte apoptosis and mitochondrial reactive oxide species production, and changed cellular metabolism from oxidative phosphorylation toward glycolysis in cardiomyocytes, thus leading to cardiomyocyte survival from ischemia and hypoxia stress. Ribo-Seq revealed that AARS2 overexpression increased pyruvate kinase M2 (PKM2) protein translation and the ratio of PKM2 dimers to tetramers that promote glycolysis. Additionally, PKM2 activator TEPP-46 reversed cardiomyocyte apoptosis and cardiac fibrosis caused by AARS2 deficiency. Thus, this study demonstrates that AARS2 plays an essential role in protecting cardiomyocytes from ischemic pressure via fine-tuning PKM2-mediated energy metabolism, and presents a novel cardiac protective AARS2-PKM2 signaling during the pathogenesis of MI.

Comments on revised version:

The authors addressed all the issues, no more comments.

---

## [Author Response]

The following is the authors’ response to the original reviews

**Public Reviews:**

**Reviewer #1 (Public Review):**
In this study, the authors introduced an essential role of AARS2 in maintaining cardiac function. They also investigated the underlying mechanism that through regulating alanine and PKM2 translation are regulated by AARS2. Accordingly, a therapeutic strategy for cardiomyopathy and MI was provided. Several points need to be addressed to make this article more comprehensive:

Thank this reviewer for the overall supports on our manuscript.

(1) Include apoptotic caspases in Figure 2B, and Figure 4 B and E as well.

This is a good point for further investigating the role of apoptosis signaling in cardiac-specific AARS2 knockout hearts. Since we are focusing on cardiomyocyte phenotypes, immunostaining on TUNEL and anti-cTnT directly evaluated the level of cardiomyocyte apoptosis, which was supported by Western blots with anti-Bcl-2 and anti-BAX of control and mutant hearts. TUNEL data accurately represents biochemical and morphological characteristics of apoptotic cells, and is more sensitive than the conventional histochemical and biochemical methods. Future studies are needed to address how apoptosis components including apoptotic caspases are involved in cardiomyocyte apoptosis in AARS2 mutant hearts.

(2) It would be better to show the change of apoptosis-related proteins upon the knocking down of AARS2 by small interfering RNA (siRNA).

Since primary culture of neonatal cardiomyocytes also contained non-cardiomyocytes, using Western blots with anti-apoptosis proteins cannot directly assess cardiomyocytes phenotypes. In this work, our data on the elevation of cTnT^+^/TUNEL^+^ cardiomyocytes and cardiac fibrosis in AARS2 mutant hearts suggest that AARS2 deficiency induced cardiomyocyte death.

(3) In Figure 5, the authors performed Mass Spectrometry to assess metabolites of homogenates. I was wondering if the change of other metabolites could be provided in the form of a heatmap.

Indeed, we assessed other metabolites by mass spectrometry as shown below, we found that overexpression of AARS2 in either transgenic mouse hearts or neonatal cardiomyocytes had no consistent changes on the level of fumarate, succinate, malate, alpha-ketoglutarate (alpha-KG), citrate, oxaloacetate (OAA), ATP, and ADP, thus suggesting that AARS2 overexpression has more specific effect on the level of lactate, pyruvate, and acetyl-CoA.

(4) The amounts of lactate should be assessed using a lactate assay kit to validate the Mass Spectrometry results.

We carried out several rounds of mass spectrometry experiments, suggesting that lactate is consistently elevated after AARS2 overexpression in neonatal cardiomyocytes as shown below. We will establish other lactate assays in future studies.

**Author response image 2. sa3fig2:** 

(5) How about the expression pattern of PKM2 before and after mouse MI. Furtherly, the correlation between AARS2 and PKM2?

Previous studies have shown that the expression level of PKM2 in mice is significantly increased after cardiac surgery at different time points, which may be related to cardiometabolic changes [1]. Our co-IP experiments showed no direct interactions between AARS2 and PKM2 (Figure 6K), while both AARS2 proteins and mRNA decreased on the 3 days (Figure 1A-B) and 7 days (Author response image 3)after myocardial infarction in mice. Thus, the level of AARS2 is reversely related to PKM2 after myocardial infarction.

**Author response image 3. sa3fig3:** 

(6) In Figure 5, how about the change of apoptosis-related proteins after administration of PKM2 activator TEPP-46?

It has been shown that TEPP-46 treatment decreased cardiomyocyte death in different models that induced cardiomyocyte apoptosis [2, 3]. We would like to refer these published works that TEPP-46 treatment improves heart function by inhibiting cardiac injury-induced cardiomyocyte death.

**Reviewer #2 (Public Review):**
Summary:The authors aimed to elucidate the role of AARS2, an alanyl-tRNA synthase, in mouse hearts, specifically its impact on cardiac function, fibrosis, apoptosis, and metabolic pathways under conditions of myocardial infarction (MI). By investigating the effects of both deletion and overexpression of AARS2 in cardiomyocytes, the study aims to determine how AARS2 influences cardiac health and survival during ischemic stress.The authors successfully achieved their aims by demonstrating the critical role of AARS2 in maintaining cardiomyocyte function under ischemic conditions. The evidence presented, including genetic manipulation results, functional assays, and mechanistic studies, robustly supports the conclusion that AARS2 facilitates cardiomyocyte survival through PKM2-mediated metabolic reprogramming. The study convincingly links AARS2 overexpression to improved cardiac outcomes post-MI, validating the proposed protective AARS2-PKM2 signaling pathway.This work may have a significant impact on the field of cardiac biology and ischemia research. By identifying AARS2 as a key player in cardiomyocyte survival and metabolic regulation, the study opens new avenues for therapeutic interventions targeting this pathway. The methods used, particularly the cardiomyocyte-specific genetic models and ribosome profiling, are valuable tools that can be employed by other researchers to investigate similar questions in cardiac physiology and pathology.Understanding the metabolic adaptations in cardiomyocytes during ischemia is crucial for developing effective treatments for MI. This study highlights the importance of metabolic flexibility and the role of specific enzymes like AARS2 in facilitating such adaptations. The identification of the AARS2-PKM2 axis adds a new layer to our understanding of cardiac metabolism, suggesting that enhancing glycolysis can be a viable strategy to protect the heart from ischemic damage.

We thank this reviewer for his/her supports on our manuscript.

Strengths:(1) Comprehensive Genetic Models: The use of cardiomyocyte-specific AARS2 knockout and overexpression mouse models allowed for precise assessment of AARS2's role in cardiac cells.(2) Functional Assays: Detailed phenotypic analyses, including measurements of cardiac function, fibrosis, and apoptosis, provided evidence for the physiological impact of AARS2 manipulation.(3) Mechanistic Insights: This study used ribosome profiling (Ribo-Seq) to uncover changes in protein translation, specifically highlighting the role of PKM2 in metabolic reprogramming.(4) Therapeutic Relevance: The use of the PKM2 activator TEPP-46 to reverse the effects of AARS2 deficiency presents a potential therapeutic avenue, underscoring the practical implications of the findings.Weaknesses:(1) Species Limitation: The study is limited to mouse and rat models, and while these are highly informative, further validation in human cells or tissues would strengthen the translational relevance.

We fully agree with this reviewer that this study is limited to mouse and rat models. It would certainly be important to address how AARS2-PKM2 is related myocardial infarction patients in the future.

(2) Temporal Dynamics: The study does not extensively address the temporal dynamics of AARS2 expression and PKM2 activity during the progression of MI and recovery, which could offer deeper insights into the timing and regulation of these processes.

Thanks for this critical point. Indeed, we found that both AARS2 proteins and mRNA decreased on 3 days (Figure 1A-B) and 7 days (Author response image 3) after myocardial infarction in mice as shown below. Others have reported PKM2 proteins increased after heart surgery in mice at different time points [1]. Thus, the level of AARS2 is reversely related to PKM2 after myocardial infarction.

**Reviewer #3 (Public Review):**
In the present study, the author revealed that cardiomyocyte-specific deletion of mouse AARS2 exhibited evident cardiomyopathy with impaired cardiac function, notable cardiac fibrosis, and cardiomyocyte apoptosis. Cardiomyocyte-specific AARS2 overexpression in mice improved cardiac function and reduced cardiac fibrosis after myocardial infarction (MI), without affecting cardiomyocyte proliferation and coronary angiogenesis. Mechanistically, AARS2 overexpression suppressed cardiomyocyte apoptosis and mitochondrial reactive oxide species production, and changed cellular metabolism from oxidative phosphorylation toward glycolysis in cardiomyocytes, thus leading to cardiomyocyte survival from ischemia and hypoxia stress. Ribo-Seq revealed that AARS2 overexpression increased pyruvate kinase M2 (PKM2) protein translation and the ratio of PKM2 dimers to tetramers that promote glycolysis. Additionally, PKM2 activator TEPP-46 reversed cardiomyocyte apoptosis and cardiac fibrosis caused by AARS2 deficiency. Thus, this study demonstrates that AARS2 plays an essential role in protecting cardiomyocytes from ischemic pressure via fine-tuning PKM2-mediated energy metabolism, and presents a novel cardiac protective AARS2-PKM2 signaling during the pathogenesis of MI. This study provides some new knowledge in the field, and there are still some questions that need to be addressed in order to better support the authors' views.

We thank this reviewer for his/her overall supports on our manuscript.

(1) WGA staining showed obvious cardiomyocyte hypertrophy in the AARS2 cKO heart. Whether AARS affects cardiac hypertrophy needs to be further tested.

WGA staining is widely used to measure the size of cardiomyocytes in the literature. Here, we found that the size of mutant cardiomyocytes increased by ~20% after AARS2 knockout. In addition, we also measured and found that the ratio of heart to body weight increased in AARS2 mutant mice compared with control siblings as shown below.

**Author response image 4. sa3fig4:** 

(2) The authors observed that AARS2 can improve myocardial infarction, and whether AARS2 has an effect on other heart diseases.

Thanks for this critical point. We agree with this reviewer that it will be important to address whether overexpression of AARS2 has cardiac protection in other heart diseases such as transverse aortic constriction in the future.

(3) Studies have shown that hypoxia conditions can lead to mitochondrial dysfunction, including abnormal division and fusion. AARS2 also affects mitochondrial division and fusion and interacts with mitochondrial proteins, including FIS and DRP1, the authors are suggested to verify.

This is a good point. Mitochondrial dysfunction occurs when cardiomyocytes are subjected to hypoxia conditions such as myocardial infarction. Our ribosome sequencing data suggested that overexpression of AARS2 had no effect on the level of FIS1 and DRP2 as shown below. We agree with this reviewer that future studies are needed to clarify potential interactions between AARS2 and FIS/DRP1 proteins.

**Author response image 5. sa3fig5:** 

(4) The authors only examined the role of AARS2 in cardiomyocytes, and fibroblasts are also an important cell type in the heart. Authors should examine the expression and function of AARS2 in fibroblasts.

We fully agree with this reviewer that AARS2 may also function in cardiac fibroblasts since it is expressed in fibroblasts and cardiomyocyte-specific AARS2 knockout led to more fibrosis after myocardial infarction, which certainly warrant future investigations.

(5) Overexpression of AARS2 can inhibit the production of mtROS, and has a protective effect on myocardial ischemia and H/ R-induced injury, and the occurrence of iron death is also closely related to ROS, whether AARS protects myocardial by regulating the occurrence of iron death?

Thank this reviewer for his/her critical point. Our current data cannot rule out whether iron-mediated death is involved in AARS2 function in cardiac protection, which warrant future investigations.

(6) Please revise the English grammar and writing style of the manuscript, spelling and grammatical errors should be excluded.

Sorry for spelling and grammatical errors. We have carefully revised this manuscript now.

(7) Recent studies have shown that a decrease in oxygen levels leads to an increase in AARS2, and lactic acid rises rapidly without being oxidized. Both of these factors inhibit oxidative phosphorylation and muscle ATP production by increasing mitochondrial lactate acylation, thereby inhibiting exercise capacity and preventing the accumulation of reactive oxygen species ROS. The key role of protein lactate acylation modification in regulating oxidative phosphorylation of mitochondria, and the importance of metabolites such as lactate regulating cell function through feedback mechanisms, i.e. cells adapt to low oxygen through metabolic regulation to reduce ROS production and oxidative damage, and therefore whether AARS2 in the heart also acts in this way.

This is an interesting question. Since overexpression of AARS2 in muscles has previously been reported to increase PDHA1 lactylation and decrease its activity [4]. Actually, we initially examined whether overexpression of AARS2 in cardiomyocytes has similar effect on PDHA1 lactylation. However, our results showed that overexpression of AARS2 had no evident increases of lactylated PDHA1 in cardiomyocytes as shown below. However, future studies are needed to explore whether other proteins lactylation by AARS2 are involved in its cardiac protection function.

**Author response image 6. sa3fig6:** 

**Reviewer #2 (Recommendations For The Authors):**
Suggestions for Improved or Additional Experiments, Data, or Analyses:(1) Validation in Human Models: It would be great if, in the future, the authors could conduct experiments with human cardiomyocytes derived from induced pluripotent stem cells (iPSCs) to validate the findings in a human context. This would strengthen the translational relevance of the results.

We fully agree with this reviewer that this study is limited to mouse and rat models. It would certainly be important to address how AARS2-PKM2 is related myocardial infarction patients and/or human iPSC-derived cardiomyocytes in the future.

(2) Broader Metabolic Analysis: To perform comprehensive metabolic profiling (e.g., metabolomics) to identify other metabolic pathways influenced by AARS2 overexpression or deficiency. This could provide a more holistic view of the metabolic changes and potential compensatory mechanisms.

As noted above, we indeed assessed other metabolites by mass spectrometry, we found that overexpression of AARS2 in either transgenic mouse hearts or neonatal cardiomyocytes had no consistent changes on the level of fumarate, succinate, malate, alpha-ketoglutarate (alpha-KG), citrate, oxaloacetic acid (OAA), ATP, and ADP, thus suggesting that AARS2 overexpression has more specific effect on the level of lactate, pyruvate, and acetyl-CoA.

(3) Temporal Dynamics: Investigate the temporal expression and activity of AARS2 and PKM2 during the progression and recovery phases of myocardial infarction. Time-course studies could elucidate the dynamics and regulatory mechanisms involved.

As noted above, we found that both AARS2 proteins and mRNA decreased on the third and seventh day after myocardial infarction in mice. Others have reported PKM2 proteins increased after heart surgery in mice at different time points [1]. Thus, the level of AARS2 is reversely related to PKM2 after myocardial infarction.

(4) Investigate Additional Pathways: Explore the involvement of other signaling pathways and tRNA synthetases that might interact with or complement the AARS2-PKM2 axis. This could uncover broader regulatory networks affecting cardiomyocyte survival and function.

Thank this reviewer for his/her critical point. This certainly warrants future investigations.

(5) Mitochondrial Function Assays: Perform detailed mitochondrial function assays, including measurements of mitochondrial respiration and membrane potential, to further elucidate the role of AARS2 in mitochondrial health and function under stress conditions.

We fully agree with this reviewer that future studies are needed to address how AARS2 is involved in mitochondrial function.

(6) Single-Cell Analysis: Utilize single-cell RNA sequencing to examine the heterogeneity in cardiomyocyte responses to AARS2 manipulation, providing insights into cell-specific adaptations and potential differential effects within the heart tissue.

We fully agree with this reviewer that it is important to address how AARS2 (cKO or overexpression) regulate cardiomyocyte heterogeneity and function in the future.

Recommendations for Improving the Writing and Presentation:(1) Visual Aids: Include more schematic diagrams to illustrate the proposed mechanisms, especially the AARS2-PKM2 signaling pathway and its impact on metabolic reprogramming. This can help readers better understand complex interactions.

Below is our working hypothesis on the role of AARS2 in cardiac protection. AARS2 deficiency caused mitochondrial dysfunction due to increasing ROS production and apoptosis while decreasing PKM2 function and glycolysis, thus leading to cardiomyopathy in mutant mice. On the other hand, overexpression of AARS2 in mice activates PKM2 and glycolysis while decreases ROS production and apoptosis, thus improving heart function after myocardial infarction.

**Author response image 7. sa3fig7:** 

(2) Discussion: Shorten the Discussion and systematically address the significance of the findings, limitations of the study, and potential future directions. This will provide a clearer narrative and context for the results.

We have now made revisions on the Discussion part to highlight the significance of this work and brief perspective of future direction.

(3) Minor corrections to the text and figures.

We have now revised the full text carefully.

(4) Typographical Errors: Carefully proofread the manuscript to correct any typographical errors and ensure consistent use of terminology and abbreviations throughout the text.

Thanks. Based on the reviewer’s suggestions, we have carefully revised the manuscript and have done proof-reading on the whole manuscript.

Availability of data, code, reagents, research ethics, or other issues:(1) Data Presentation: Ensure that all graphs and charts are clearly labeled with appropriate units, scales, and legends. Use color schemes that are accessible to color-blind readers.

We followed these rules to present the data.

(2) Supplementary Information: Provide detailed supplementary information, including raw data, experimental protocols, and analysis scripts, to enhance the reproducibility of the study.

We provided the raw data, experimental protocols, and analysis scripts in the manuscript.

(3) Data and Code Availability. Data Sharing: Authors should ensure that all raw data, processed data, and relevant metadata are deposited in publicly accessible repositories. Provide clear instructions on how to access these data. Code Availability: Make all analysis code available in a public repository, such as GitHub, with adequate documentation to allow other researchers to replicate the analyses.

We have deposited RNA-Seq data at ArrayExpress (E-MTAB-13767). We have also uploaded the original data in the supplementary file.

(4) Research Ethics and Compliance. Ethics Statement: Include a detailed statement on the ethical approval obtained for animal experiments, specifying the institution and ethical review board that granted approval. Conflict of Interest: Clearly state any potential conflicts of interest and funding sources that supported the research to ensure transparency.

Thanks. In the manuscript we made an ethical statement, stating conflicts of interest and sources of funding.

References:

(1) Y. Tang, M. Feng, Y. Su, T. Ma, H. Zhang, H. Wu, X. Wang, S. Shi, Y. Zhang, Y. Xu, S. Hu, K. Wei, D. Xu, Jmjd4 Facilitates Pkm2 Degradation in Cardiomyocytes and Is Protective Against Dilated Cardiomyopathy, Circulation, 147 (2023) 1684-1704.

(2) L. Guo, L. Wang, G. Qin, J. Zhang, J. Peng, L. Li, X. Chen, D. Wang, J. Qiu, E. Wang, M-type pyruvate kinase 2 (PKM2) tetramerization alleviates the progression of right ventricle failure by regulating oxidative stress and mitochondrial dynamics, Journal of translational medicine, 21 (2023) 888.

(3) B. Saleme, V. Gurtu, Y. Zhang, A. Kinnaird, A.E. Boukouris, K. Gopal, J.R. Ussher, G. Sutendra, Tissue-specific regulation of p53 by PKM2 is redox dependent and provides a therapeutic target for anthracycline-induced cardiotoxicity, Science translational medicine, 11 (2019).

(4) Y. Mao, J. Zhang, Q. Zhou, X. He, Z. Zheng, Y. Wei, K. Zhou, Y. Lin, H. Yu, H. Zhang, Y. Zhou, P. Lin, B. Wu, Y. Yuan, J. Zhao, W. Xu, S. Zhao, Hypoxia induces mitochondrial protein lactylation to limit oxidative phosphorylation, Cell research, 34 (2024) 13-30.